## PROCEEDINGS A

differential equations, applied mathematics, artificial intelligence

system identification, machine learning, sparse regression, dynamical systems

**Author for correspondence:**
Pawan Goyal
e-mail: goyalp@mpi-magdeburg.mpg.de

# Discovery of nonlinear dynamical systems using a Runge–Kutta inspired dictionary-based sparse regression approach

## Pawan Goyal and Peter Benner

Max Planck Institute for Dynamics of Complex Technical Systems, Standtorstraße 1, 39106 Magdeburg, Germany

PG, 0000-0003-3072-7780; PB, 0000-0003-3362-4103

In this work, we blend machine learning and dictionary-based learning with numerical analysis tools to discover differential equations from noisy and sparsely sampled measurement data of time-dependent processes. We use the fact that given a dictionary containing large candidate nonlinear functions, dynamical models can often be described by a few appropriately chosen basis functions. As a result, we obtain parsimonious models that can be better interpreted by practitioners, and potentially generalize better beyond the sampling regime than black-box modelling. In this work, we integrate a numerical integration framework with dictionary learning that yields differential equations without requiring or approximating derivative information at any stage. Hence, it is utterly effective for corrupted and sparsely sampled data. We discuss its extension to governing equations, containing rational nonlinearities that typically appear in biological networks. Moreover, we generalized the method to governing equations subject to parameter variations and externally controlled inputs. We demonstrate the efficiency of the method to discover a number of diverse differential equations using noisy measurements, including a model describing neural dynamics, chaotic Lorenz model, Michaelis–Menten kinetics and a parameterized Hopf normal form.

# 1. Introduction

Data-driven discovery of dynamical models has recently drawn significant attention as there have been revolutionary breakthroughs in data science and machine learning [1,2]. With the increasing ease of data availability and advances in machine learning, we can analyse data and identify patterns to uncover dynamical models that faithfully describe the underlying dynamical behaviour. Although inference of dynamical models has been intensively studied in the literature, drawing conclusions and interpretations still remains tedious. Moreover, extrapolation and generalization of models are limited beyond the training regime.

The area of identifying models using data is often referred to as system identification. For linear systems, there is an extensive collection of approaches [3,4]. However, despite several decades of research on learning nonlinear systems [5–8], the field is still far away from being as mature as that for linear systems. Inferring nonlinear systems often requires *a priori* model hypothesis by practitioners. A compelling breakthrough towards discovering nonlinear governing equations appeared in [9,10], where an approach based on genetic programming or symbolic regression is developed to identify nonlinear models using measurement data. It provides parsimonious models that accomplish a long-standing desire for the engineering community. A parsimonious model is determined by examining the Pareto front that discloses a trade-off between the identified model's complexity and accuracy. In a similar spirit, there have been efforts to develop sparsity promoting approaches to discover nonlinear dynamical systems [11–15]. It is often observed that the dynamics of physical processes can be given by collecting a few nonlinear feature candidates from a high-dimensional nonlinear function space, referred to as a feature dictionary. These sparsity-promoting methods are able to discover models that are parsimonious, which in some situations can lead to better interpretability than black-box models. For motivation, we take an example from [14], where using data for fluid flow dynamics behind a cylinder, it is shown that one can obtain a model, describing the dynamics on-attractor and off-attractor and characterizing a slow parabolic manifold. Fluid dynamics practitioners can well interpret this model. Another example may come from biological modelling, where parsimonious models can describe how a species affects the dynamics of other species. Hence, the approach to discovering sparse models using dictionary learning can be interpreted in this way.

Significant progress in solving sparse regression problems [16–18] and in compressed sensing [19–22] supports the development of these approaches. Although all these methods have gained much popularity, the success largely depends on the feature candidates included in the dictionary and the ability to approximate the derivative information using measurement data accurately. A derivative approximation using sparsely sampled and noisy measurements imposes a tough challenge though there are approaches to deal with noise, e.g. [23]. We also highlight additional directions explored in the literature to discover nonlinear governing equations, which include discovery of models using time-series data [8], automated inference of dynamics [9,24,25] and equation-free modelling [13,26,27].

In this work, we re-conceptualize the problem of discovering nonlinear differential equations by blending sparse identification with a classical numerical integration tool. We focus on a widely known integration scheme, namely the classical fourth-order *Runge–Kutta* [28] method, noting that any other explicit high-order integration scheme, e.g. 3/8-rule fourth Runge–Kutta method or the ideal of neural ODEs proposed in [29] incorporating any numerical integrator. In contrast to previously studied sparse identification approaches, e.g. [9,11,14], our approach does not require direct access or approximation of temporal gradient information. Therefore, we do not commit errors due to a gradient approximation. The approach becomes an attractive choice when the collected measurement data are sparsely sampled and corrupted with noise.

However, we mention that using numerical integration schemes in the course of learning dynamics has a relatively long history. The work goes back to [30,31], where the fourth-order Runge–Kutta scheme is coupled with neural networks to learn a function, describing the underlying vector field. In recent times, making use of numerical integration schemes with

neural networks has again received attention and has been studied from the perspective of dynamical modelling, e.g. [32–34]. We particularly emphasize the work [34] that also uses a similar concept to learn dynamical systems using noisy measurements; precisely, it realizes the decoupling of noise and the underlying truth by enforcing a time-stepping integration scheme. As a result, one may obtain a denoised signal and the dynamical models describing the underlying vector field. Based on this, we have discussed an approach in [35] to learn dynamical models from noisy measurements using time-stepping schemes combined with neural networks that can handle missing data as well, which is not possible in the approach discussed in [34]. Recently, neural ODEs [29] have gained popularity to learn dynamical systems that show how to fuse any numerical integration efficiently in the course of learning models. Despite all these aforementioned methods being very general in the sense that they do not require any prior assumptions about the underlying system or structure of dynamical models, they are often black-box models; thus, interpretability and generalization of these models is unclear.

In this work, we also discuss an essential class of dynamical models that typically explains the dynamics of biological networks. It is also shown that regulatory and metabolic networks are sparse in nature, i.e. not all components influence each other. Furthermore, such dynamical models are often given by rational nonlinear functions. Consequently, the classical dictionary-based sparse identification ideology is not applicable as building all possible rational feature candidates is infeasible. To deal with this, the authors in [36] have recast the problem as finding the sparsest vector in a given null space. However, computing a null space using corrupted measurement data is a non-trivial task though there is some work in this direction [37]. Here, we instead characterize identifying rational functions as a fraction of two functions, where each function is identified using dictionary learning. Hence, we inherently retain the primary principle of sparse identification in the course of discovering models. In addition to these, we discuss the case where a dictionary contains parameterized candidates, e.g. $e^{\alpha x}$, where $x$ is the dependent variable, and $\alpha$ is an unknown parameter. We extend our discussion to parametric and controlled dynamic processes. The organization of the paper is as follows. In §2, we briefly recap the classical fourth-order Runge–Kutta method for the integration of ordinary differential equations. After that, we propose a methodology to discover differential equations by synthesizing the integration scheme with sparse identification. Furthermore, since the method involves solving nonlinear and non-convex optimization problems that promote sparse solutions, §3 discusses algorithms inspired by a sparse-regression approach in [14,18]. In §4, we examine a number of extensions to other classes of models, e.g. when the governing equations are given by a fraction of two functions and involve model parameters and external control inputs. In the subsequent section, we illustrate the efficiency of the proposed methods by discovering a broad variety of benchmark examples, namely the chaotic Lorenz model, Fitz–Hugh Nagumo (FHN) models, Michaelis–Menten kinetics and parameterized Hopf normal norm. We extensively study the performance of the proposed approach even under noisy measurements and compare it to the approach proposed in [14]. We conclude the paper with a summary and high-priority research directions.

## 2. Discovering nonlinear governing equations using a Runge–Kutta inspired sparse identification

In this section, we describe our approach to discovering nonlinear governing equations using sparsely sampled measurement data. These may be corrupted using experimental and/or sensor noise. We establish approaches by combining a numerical integration method and dictionary-based learning. So, we develop methodologies that allow us to discover nonlinear differential equations without the explicit need for derivative information, unlike the approach proposed in e.g. [11,14,25]. In this work, we use the widely employed approach to integrate differential equations, namely the classical *fourth-order Runge–Kutta* (RK4) method, which is briefly outlined next.

## (a) Fourth-order Runge–Kutta method

The RK4 scheme is a widely used method to solve initial value problems. Let us consider the following initial value problem:

$$\dot{\mathbf{x}}(t) = \mathbf{f}(\mathbf{x}(t)) \quad \text{and} \quad \mathbf{x}(t_0) = \mathbf{x}_0, \tag{2.1}$$

where $\mathbf{x}(t) := [\mathbf{x}_1(t), \mathbf{x}_2(t), \dots, \mathbf{x}_n(t)]$ with $\mathbf{x}_j(t)$ being the $j$th element of the vector $\mathbf{x}(t)$, and the function $: \mathbb{R}^n \to \mathbb{R}^n$ defines its vector field. Assume that we aim at predicting $\mathbf{x}(t_{k+1})$ for a given $\mathbf{x}(t_k)$, where $k \in \{0, 1, \dots, \mathcal{N}\}$. Then, using the RK4 scheme, $\mathbf{x}(t_{k+1})$ can be given as a weighted sum of four increments, which are the product of the time-step and vector field information $\mathbf{f}(\cdot)$ at specific locations. Precisely, it is given as

$$\mathbf{x}(t_{k+1}) \approx \mathbf{x}(t_k) + \frac{1}{6} h_k (\mathbf{k}_1 + 2 \cdot \mathbf{k}_2 + 2 \cdot \mathbf{k}_3 + \mathbf{k}_4), \quad h_k = t_{k+1} - t_k, \tag{2.2}$$

where

$$\mathbf{k}_1 = \mathbf{f}(\mathbf{x}(t_k)), \quad \mathbf{k}_2 = \mathbf{f}(\mathbf{x}(t_k) + h_k \frac{\mathbf{k}_1}{2}), \quad \mathbf{k}_3 = \mathbf{f}(\mathbf{x}(t_k) + h_k \frac{\mathbf{k}_2}{2}) \quad \text{and} \quad \mathbf{k}_4 = \mathbf{f}(\mathbf{x}(t_k) + h_k \mathbf{k}_3).$$

The RK4 scheme as a network is illustrated in figure 1a. The local integration error due to the RK4 scheme is in $\mathcal{O}(h_k^5)$; hence, the approach is very accurate for small time-steps. Furthermore, if we integrate equation (2.1) from $t_0$ to $t_f$, we can take $\mathcal{N}$ steps with time-steps $h_k, k \in \{1, \dots, \mathcal{N}\}$ so that $t_f = t_0 + \sum_{i=0}^{\mathcal{N}} h_k$. In the rest of the paper, we use the short-hand notation $\mathcal{F}_{\text{RK4}}(\mathbf{f}, \mathbf{x}(t_k), h_k)$, i.e. for the step in (2.2)

$$\mathbf{x}(t_{k+1}) = \mathbf{x}(t_k + h_k) \approx \mathcal{F}_{\text{RK4}}(\mathbf{f}, \mathbf{x}(t_k), h_k). \tag{2.3}$$

Lastly, we stress the point that the RK4 scheme readily handles integration backward in time, meaning that $h_k$ in (2.2) can also be negative. Hence, we can predict both $\mathbf{x}(t_{k+1})$ and $\mathbf{x}(t_{k-1})$ using $\mathbf{x}(t_k)$ very accurately using the RK4 scheme.

## (b) Discovering nonlinear dynamical systems

Next, we develop a RK4-inspired sparse identification approach to discover governing equations. Precisely, we aim at disclosing the most parsimonious representation of the vector field $\mathbf{f}(\mathbf{x}(t))$ in (2.1) using only a time-history of $\mathbf{x}(t)$. Assume that the data are sampled at the time instances $\{t_0, \dots, t_{\mathcal{N}}\}$, and let us define time-steps $h_k := t_{k+1} - t_k$. Furthermore, for simplicity of notation, we assume that the data follows RK4 exactly, but of course, the method is not limited to this, as we see in our numerical experiments. Consequently, we form two data matrices

$$\mathbf{X} := \begin{bmatrix} \mathbf{x}(t_1) \\ \mathbf{x}(t_2) \\ \vdots \\ \mathbf{x}(t_{\mathcal{N}}) \end{bmatrix} = \begin{bmatrix} \mathbf{x}_1(t_1) & \mathbf{x}_2(t_1) & \cdots & \mathbf{x}_n(t_1) \\ \mathbf{x}_1(t_2) & \mathbf{x}_2(t_2) & \cdots & \mathbf{x}_n(t_2) \\ \vdots & \vdots & \ddots & \vdots \\ \mathbf{x}_1(t_{\mathcal{N}}) & \mathbf{x}_2(t_{\mathcal{N}}) & \cdots & \mathbf{x}_n(t_{\mathcal{N}}) \end{bmatrix} \quad \text{and} \quad \mathbf{X}_{\mathcal{F}}(\mathbf{f}) := \begin{bmatrix} \mathcal{F}_{\text{RK4}}(\mathbf{f}, \mathbf{x}(t_0), h_1) \\ \mathcal{F}_{\text{RK4}}(\mathbf{f}, \mathbf{x}(t_1), h_2) \\ \vdots \\ \mathcal{F}_{\text{RK4}}(\mathbf{f}, \mathbf{x}(t_{\mathcal{N}-1}), h_{\mathcal{N}}) \end{bmatrix}. \tag{2.4}$$

The next important ingredient to sparse identification is the construction of a huge symbolic dictionary $\boldsymbol{\Phi}$, containing potential nonlinear features. We assume that the function $\mathbf{f}(\cdot)$ can be given by a linear combination of few terms from the dictionary. For example, one can consider a dictionary containing polynomial, exponential and trigonometric functions, which, for any given vector $\mathbf{v} := [\mathbf{v}_1, \dots, \mathbf{v}_n]$ can be given as

$$\boldsymbol{\Phi}(\mathbf{v}) = \left[ 1, \mathbf{v}, \mathbf{v}^{\mathcal{P}_2}, \mathbf{v}^{\mathcal{P}_3}, \dots, \mathbf{e}^{-\mathbf{v}}, \mathbf{e}^{-2\mathbf{v}}, \dots, \sin(\mathbf{v}), \cos(\mathbf{v}), \dots \right] \tag{2.5}$$

in which $\mathbf{v}^{\mathcal{P}_i}, i \in \{2, 3\}$, denote high-order polynomials, e.g. $\mathbf{v}^{\mathcal{P}_2}$ contains all possible degree-2 polynomials of elements of $\mathbf{v}$ as

$$\mathbf{v}^{\mathcal{P}_2} = \left[ \mathbf{v}_1^2, \mathbf{v}_1 \mathbf{v}_2, \dots, \mathbf{v}_2^2, \mathbf{v}_2 \mathbf{v}_3, \dots, \mathbf{v}_n^2 \right]. \tag{2.6}$$

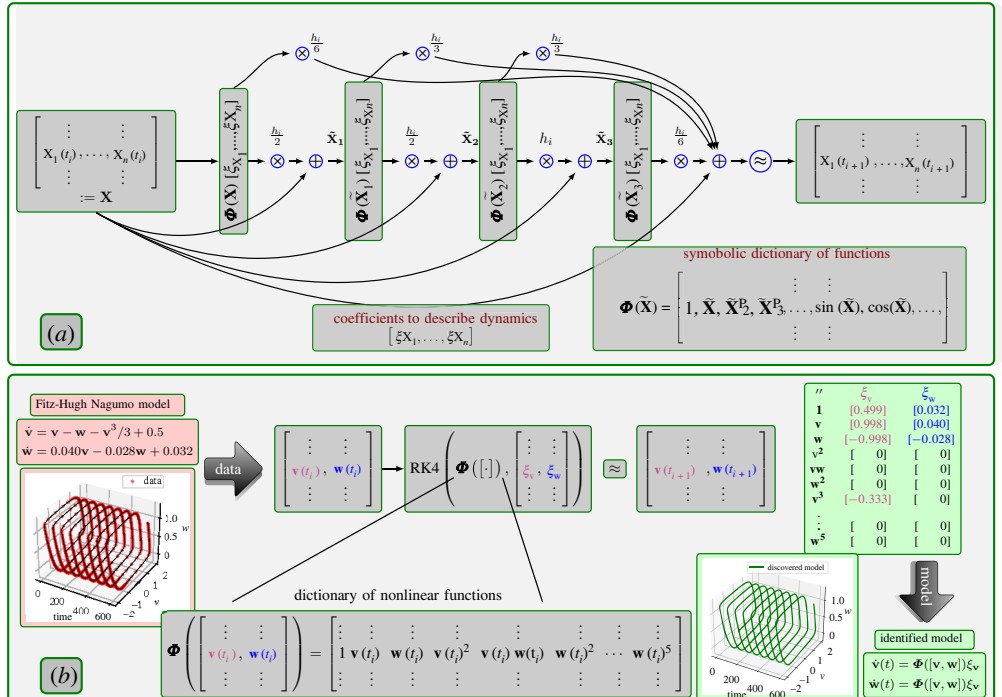

**Figure 1.** In (*a*), we show the RK4 scheme to predict variables at the next time-step as a network. It resembles a residual-type network with skip connections (e.g. [38,39]). In (*b*), we present a systematic illustration of the RK4-SINDy approach to discover governing equations using the Fitz–Hugh Nagumo model. In the first step, we collect a time history of variables $\mathbf{v}(t)$ and $\mathbf{w}(t)$. Next, we build a symbolic feature dictionary $\boldsymbol{\Phi}$, containing potential features. This is followed by solving a nonlinear sparse regression problem to pick the right features from the dictionary (encoded in sparse vectors $\xi_\mathbf{v}$ and $\xi_\mathbf{w}$). Here, we presume that variables at the next time-steps are given by following the RK4 scheme. The non-zero entries in vectors $\xi_\mathbf{v}$ and $\xi_\mathbf{w}$ determine the right-hand side of the differential equations. As shown, we pick the right features from the dictionary upon solving the optimization problem, and corresponding coefficients are 0.1% accurate. (Online version in colour.)

Each element in the dictionary $\boldsymbol{\Phi}$ is a potential candidate to describe the function $\mathbf{f}$. Moreover, depending on the application, one may use empirical or expert knowledge to construct a meaningful feature dictionary.

Having an extensive dictionary, one has many choices of candidates. However, our goal is to choose as few candidates as possible, describing the nonlinear function $\mathbf{f}$ in (2.1). Hence, we set up a sparsity-promoting optimization problem to pick a few candidate functions from the dictionary, e.g.

$$\mathbf{f}_k(\mathbf{x}(t)) = \boldsymbol{\Phi}(\mathbf{x}(t))\boldsymbol{\xi}_k, \qquad (2.7)$$

where $\mathbf{f}_k : \mathbb{R}^n \to \mathbb{R}$ is the $k$th element of $\mathbf{f}$, and $\boldsymbol{\xi}_k \in \mathbb{R}^m$ a sparse vector with $m$ the total number of features in the dictionary $\boldsymbol{\Phi}$; hence, selecting appropriate candidates from the dictionary determines the governing equations. As a result, we can write the function $\mathbf{f}(\cdot)$ in (2.1) as follows:

$$\mathbf{f}(\mathbf{x}) = \left[ \mathbf{f}_1(\mathbf{x}), \mathbf{f}_2(\mathbf{x}), \ldots, \mathbf{f}_n(\mathbf{x}) \right] = \left[ \boldsymbol{\Phi}(\mathbf{x})\boldsymbol{\xi}_1, \boldsymbol{\Phi}(\mathbf{x})\boldsymbol{\xi}_2, \ldots, \boldsymbol{\Phi}(\mathbf{x})\boldsymbol{\xi}_n \right] = \boldsymbol{\Phi}(\mathbf{x})\boldsymbol{\Xi}, \qquad (2.8)$$

where $\boldsymbol{\Xi} = [\boldsymbol{\xi}_1, \ldots, \boldsymbol{\xi}_n]$. This allows us to articulate our optimization problem that aims at discovering governing equations; that is to find the sparsest $\boldsymbol{\Xi}$, satisfying

$$\mathbf{X} = \mathbf{X}_{\mathcal{F}}(\mathbf{f}), \quad \text{where } \mathbf{f}(\mathbf{x}) = \boldsymbol{\Phi}(\mathbf{x})\boldsymbol{\Xi}. \qquad (2.9)$$

Once we identify $\boldsymbol{\Xi}$ or $\{\boldsymbol{\xi}_1, \ldots, \boldsymbol{\xi}_n\}$, the dynamical model is given as

$$\left[ \mathbf{x}_1(t), \mathbf{x}_2(t), \ldots, \mathbf{x}_n(t) \right] = \left[ \boldsymbol{\Phi}(\mathbf{x})\boldsymbol{\xi}_1, \boldsymbol{\Phi}(\mathbf{x})\boldsymbol{\xi}_2, \ldots, \boldsymbol{\Phi}(\mathbf{x})\boldsymbol{\xi}_n \right].$$

We refer to the proposed approach as Runge–Kutta inspired sparse identification for dynamical systems (RK4-SINDy). We depict all the essential steps for RK4-SINDy to discover governing equations in figure 1 through the FHN model (details of the model are provided later).

We take the opportunity to stress the imperative advantages of RK4-SINDy. That is, to discover nonlinear differential equations, we do not require derivative information of $\mathbf{x}(t)$ at any step. We only hypothesize that the vector field can be given by selecting appropriate features from a dictionary containing a vast number of possible nonlinear features. Consequently, we expect to discover good quality models when data are sparsely collected or are corrupted, and this is what we manifest in our results in §5. The approach is also appealing when data is collected at irregular time-steps.

When the data are corrupted with noise or do not follow RK4 exactly, we may need to regularize the above optimization problem. Since the $l_1$-regularization promotes sparsity in the solution, one can solve an $l_1$-regularized optimization problem

$$\min_{\boldsymbol{\Xi}} ||\mathbf{X} - \mathbf{X}_{\mathcal{F}}(\boldsymbol{\Phi}(\cdot)\boldsymbol{\Xi})|| + \alpha||\boldsymbol{\Xi}||_{l_1}, \tag{2.10}$$

where $\alpha$ is a regularizing parameter.

As discussed in §a, the RK4 scheme can accurately predict both $\mathbf{x}(t_{i+1})$ and $\mathbf{x}(t_{i-1})$ using $\mathbf{x}(t_i)$. Therefore, the following also holds

$$\mathbf{X}^\flat = \mathbf{X}_{\mathcal{F}}^\flat(\mathbf{f}),$$

where

$$\mathbf{X}^\flat := \begin{bmatrix} \mathbf{x}(t_0) \\ \mathbf{x}(t_1) \\ \vdots \\ \mathbf{x}(t_{\mathcal{N}-1}) \end{bmatrix} = \begin{bmatrix} \mathbf{x}_1(t_0) & \mathbf{x}_2(t_0) & \cdots & \mathbf{x}_n(t_0) \\ \mathbf{x}_1(t_1) & \mathbf{x}_2(t_1) & \cdots & \mathbf{x}_n(t_2) \\ \vdots & \vdots & \ddots & \vdots \\ \mathbf{x}_1(t_{\mathcal{N}-1}) & \mathbf{x}_2(t_{\mathcal{N}-1}) & \cdots & \mathbf{x}_n(t_{\mathcal{N}-1}) \end{bmatrix} \quad \mathbf{X}_{\mathcal{F}}^\flat(\mathbf{f}) := \begin{bmatrix} \mathcal{F}_{\mathsf{RK4}}(\mathbf{f}, \mathbf{x}(t_1), -h_1) \\ \mathcal{F}_{\mathsf{RK4}}(\mathbf{f}, \mathbf{x}(t_2), -h_2) \\ \vdots \\ \mathcal{F}_{\mathsf{RK4}}(\mathbf{f}, \mathbf{x}(t_{\mathcal{N}}), -h_{\mathcal{N}}) \end{bmatrix}.$$

Therefore, we can have a more involved optimization by including both forward and backward predictions in time. This helps particularly in the presence of noisy measurement data. But, on the other hand, it would make the optimization problems yielding the sparse vectors $\boldsymbol{\xi}_i$'s harder to solve. In the following subsection, we discuss an efficient procedure to solve the optimization problem (2.9).

## 3. Algorithms to solve nonlinear sparse regression problems

Several methodologies exist to solve linear optimization problems that yield a sparse solution, e.g. LASSO [16,18]. However, the optimization problem (2.10) is nonlinear and likely non-convex. There are some developments in solving sparsity-constrained nonlinear optimization problems; e.g. [40, 41]. Though these methods enjoy many nice theoretical properties, they typically require *a priori* the maximum number of non-zero elements in the solutions, which is often unknown. Also, they are computationally demanding. Here, we propose two simple gradient-based sequential thresholding schemes, similar to the one discussed in [14] for linear problems. In these schemes, we first solve the nonlinear optimization problem (2.10) using a (stochastic) gradient descent method to obtain $\boldsymbol{\Xi}_1$, followed by applying a thresholding to $\boldsymbol{\Xi}_1$.

## (a) Fix cut-off thresholding

In the first approach, we define a cut-off value $\lambda$ and set all the coefficients smaller than $\lambda$ to zero. We then update the remaining non-zero coefficients by solving the optimization problem (2.10) again, followed by employing the thresholding. We repeat the procedure until all the non-zero

---

**Algorithm 1**. Fix Cutoff Thresholding Procedure.

---

**Input:** Measurement data $\{\mathbf{x}(t_0), \mathbf{x}(t_1), \ldots, \mathbf{x}(t_\mathcal{N})\}$ and the cutoff parameter $\lambda$.

1: Solve the optimization problem (2.10) to get $\varXi$.
2: `small_idx` $= (|\varXi| < \lambda)$.                ▷ Determine indices at which coefficients are less than $\lambda$
3: `Err` $= \|\varXi\,(\text{small\_idx})\|$.
4: **while** `Err` $\neq 0$ **do**
5:    Update $\varXi$ by solving the problem (2.10) with the constraint $\varXi\,(\text{small\_idx}) = 0$.
6:    `small_idx` $= (|\varXi| < \lambda)$.         ▷ Determine indices at which coefficients are less than $\lambda$
7:    `Err` $= \|\varXi\,(\text{small\_idx})\|$.

**Output:** The sparse $\varXi$ that picks the right features from the dictionary.

---

coefficients are equal to or larger than $\lambda$. This procedure is efficient as the current value of non-zero coefficients can be used as an initial guess for the next iteration, and the optimal $\varXi$ can be found with little computational effort. Note that the cut-off parameter $\lambda$ is important to obtain a suited sparse solution, but it can be found using the concept of cross-validation. We sketch the discussed procedure in algorithm 1.

For the iterative thresholding algorithm proposed for the sparse regression in [14], an analysis of the iterative thresholding algorithm is conducted in [42], showing a rapid convergence of the algorithm. In contrast to the algorithm in [14], algorithm 1 is more complex, and the underlying optimization problem is non-convex; thus, a thorough study of its convergence is out of the scope of this paper. However, we here mention that a rapid convergence of algorithm 1 is observed in numerical experiments, but its analysis will be an important topic for future research.

We also note that algorithm 1 always terminates as either the number of indices set to zero is increased (which terminates when the dictionary is exhausted), or the error criterion is satisfied. But the question remains as to whether the algorithm will converge to the correct sparse solution. A remedy to this can be to use the rationale of an ensemble, proposed in [43] to build an ensemble of sparse models. It can provide statistical quantities for the feature candidates in the dictionary. Based on these, we can construct a final sparse model based on statistical tools such as the $p$-values.

## (b) Iterative cut-off thresholding

In the fix cut-off thresholding approach, we need to pre-define the cut-off value for thresholding. A suitable value of it needs to be found by an iterative procedure. In our empirical observations, applying fix thresholding at each iteration does not yield the most sparse solution in many instances. To circumvent this, we propose an iterative way of thresholding, as follows. In the first step, we solve the optimization problem (2.10) for $\varXi$. Then, we determine the smallest non-zero coefficients of $|\varXi|$ followed by setting all the coefficients smaller than this to zero. Like in the previous approach, we update the remaining non-zero coefficients by solving the optimization problem (2.10). We repeat the step of finding the smallest non-zero coefficient of the updated $|\varXi|$ and setting it to zero. We iterate the procedure until the loss of data fidelity is less than a given tolerance. Visually, it can be anticipated using the curve between the data-fitting and number of non-zero elements in $\varXi$, which typically exhibit an *elbow*-type curve. This approach is close to the *backward stepwise selection* approach used in machine learning for feature selection, e.g. [17]. We sketch the step of the procedure in algorithm 2. We shall see the use of this algorithm in our results section (see §d).

We note that the successive iterations converge faster to the optimal value after the first thresholding as we choose the coefficients after applying thresholding as the initial guess. Moreover, in our experiments, we observe that this thresholding approach yields better results,

**Algorithm 2**. Iterative Cutoff Thresholding Procedure.

---

**Input:** Measurement data $\{\mathbf{x}(t_0), \mathbf{x}(t_1), \ldots, \mathbf{x}(t_\mathcal{N})\}$.

1: Construct $\mathbf{X}$ using measurement data as in (2.4).
2: Solve the optimization problem (2.10) to get $\boldsymbol{\Xi}$.
3: $\mathcal{E} := \|\mathbf{X} - \mathbf{X}_\mathcal{F}(\boldsymbol{\Phi}(\cdot)\boldsymbol{\Xi})\|$, where $\mathbf{X}_\mathcal{F}$ is defined in (2.4).
4: **while** $\mathcal{E} \leq \texttt{tol}$ **do**
5:     Determine the smallest non-zero coefficient of $\texttt{abs}(\boldsymbol{\Xi})$, denoted by $\lambda_{\texttt{small}}$.
6:     $\texttt{small\_idx} = (|\boldsymbol{\Xi}| < \lambda_{\texttt{small}})$     ▷ Determine indices at which coefficients are less than $\lambda$
7:     Update $\boldsymbol{\Xi}$ by solving the optimization problem (2.10) with the constraint $\boldsymbol{\Xi}(\texttt{small\_idx}) = 0$.
8:     $\mathcal{E} := \|\mathbf{X} - \mathbf{X}_\mathcal{F}(\boldsymbol{\Phi}(\cdot)\boldsymbol{\Xi})\|$.

**Output:** The sparse $\boldsymbol{\Xi}$ that picks right features from the dictionary.

---

particularly when data are corrupted with noise. However, it may be computationally more expensive than the fixed cut-off thresholding approach as it may need more iterations to converge. Therefore, an efficient approach combining fixed and iterative thresholding approaches is a worthy future research direction.

# 4. A number of possible extensions

In this section, we discuss several extensions of the methodology proposed in §2, generalizing to a large class of problems. First, we discuss the discovery of governing differential equations given by a fraction of two functions. Next, we investigate the case in which a symbolic dictionary is parameterized. It is of particular interest when governing equations expected to have candidate features, e.g. $\mathbf{e}^{\alpha \mathbf{x}(t)}$, where $\alpha$ is unknown. We further extend our discussion to parameterized and externally controlled governing equations.

## (a) Governing equations as a fraction of two functions

There are many instances where the governing equations are given as a fraction of two nonlinear functions. Such equations frequently appear in the modelling of biological networks. For simplicity, we here examine a scalar problem; however, the extension to multi-dimensional cases readily follows. Consider governing equations of the form

$$\dot{\mathbf{x}}(t) = \frac{\mathbf{g}(\mathbf{x})}{1 + \mathbf{h}(\mathbf{x})}, \tag{4.1}$$

where $\mathbf{g}(\mathbf{x}) : \mathbb{R} \to \mathbb{R}$ and $\mathbf{h}(\mathbf{x}) : \mathbb{R} \to \mathbb{R}$ are continuous nonlinear functions. Here again, the observation is that the functions $\mathbf{g}(\cdot)$ and $\mathbf{h}(\cdot)$ can be given as linear combinations of a few terms from corresponding dictionaries. Hence, we can cast the problem of identifying the model (4.1) as a dictionary-based discovery of governing equations. Let us consider two symbolic dictionaries

$$\boldsymbol{\Phi}^{(\mathbf{g})}(\mathbf{x}) = \left[ 1, \mathbf{x}, \mathbf{x}^2, \mathbf{x}^3, \ldots, \sin(\mathbf{x}), \cos(\mathbf{x}), \sin(\mathbf{x}^2), \cos(\mathbf{x}^2), \sin(\mathbf{x}^3), \sin(\mathbf{x}^3), \ldots \right] \tag{4.2}$$

and

$$\boldsymbol{\Phi}^{(\mathbf{h})}(\mathbf{x}) = \left[ \mathbf{x}, \mathbf{x}^2, \mathbf{x}^3, \ldots, \sin(\mathbf{x}), \cos(\mathbf{x}), \sin(\mathbf{x}^2), \cos(\mathbf{x}^2), \sin(\mathbf{x}^3), \sin(\mathbf{x}^3), \ldots \right]. \tag{4.3}$$

Consequently, the functions $\mathbf{g}(\cdot)$ and $\mathbf{h}(\cdot)$ can be given by

$$\mathbf{g}(\mathbf{x}) = \boldsymbol{\Phi}^{(\mathbf{g})}(\mathbf{x})\boldsymbol{\xi}_{\mathbf{g}} \tag{4.4}$$

and

$$\mathbf{h}(\mathbf{x}) = \boldsymbol{\Phi}^{(\mathbf{h})}(\mathbf{x})\boldsymbol{\xi}_{\mathbf{h}}, \tag{4.5}$$

where $\xi_g$ and $\xi_h$ are sparse vectors. Then, we can readily apply the framework discussed in the previous section by assuming $f(x) := g(x)/(1 + h(x))$ in (2.1). We can determine sparse coefficients $\xi_g$ and $\xi_h$ by employing the thresholding concepts presented in algorithms 1 and 2. These are possible because the algorithms are gradient-based and we only need to compute gradients with respect to $\xi_g$ and $\xi_h$.

Furthermore, it is worthwhile to consider governing equations of the form

$$\dot{x}(t) = k(x) + \frac{g(x)}{1 + h(x)}. \tag{4.6}$$

Indeed, the model (4.6) can be rewritten in the form considered in (4.1). But it is rather efficient to consider the form (4.6). We illustrate it with the following example:

$$\dot{x}(t) = -x(t) - \frac{x(t)}{1 + x(t)}, \tag{4.7}$$

which fits to the form considered in (4.6). In this case, all nonlinear functions $k(\cdot), g(\cdot)$ and $h(\cdot)$ are degree-1 polynomials. On the other hand, if the model (4.7) is written in the form (4.1), then we have

$$\dot{x}(t) = \frac{-1 - x(t) - x(t)^2}{1 + x(t)}. \tag{4.8}$$

Thus, the nonlinear functions $g(\cdot)$ and $h(\cdot)$ in (4.1) are of degrees 2 and 1, respectively. This gives a hint that if we aim at learning governing equations using sparse identification, it might be efficient to consider the form (4.6) due to a smaller size of the necessary dictionary. It becomes even more adequate in multi-dimensional differential equations. To discover a dynamical model of the form (4.6), we extend the idea of learning nonlinear functions using dictionaries. Let us construct three dictionaries as follows:

$$\Phi^{(k)}(x) = \left[1, x, x^2, x^3, \ldots, \sin(x), \cos(x), \sin(x^2), \cos(x^2), \sin(x^3), \sin(x^3), \ldots\right], \tag{4.9}$$

$$\Phi^{(g)}(x) = \left[1, x, x^2, x^3, \ldots, \sin(x), \cos(x), \sin(x^2), \cos(x^2), \sin(x^3), \sin(x^3), \ldots\right] \tag{4.10}$$

and

$$\Phi^{(h)}(x) = \left[x, x^2, x^3, \ldots, \sin(x), \cos(x), \sin(x^2), \cos(x^2), \sin(x^3), \sin(x^3), \ldots\right]. \tag{4.11}$$

Then, we believe that the nonlinear functions in (4.6) can be given as a sparse linear combination of the dictionaries, i.e.

$$k(x) = \Phi^{(k)}(x)\xi_k, \quad g(x) = \Phi^{(g)}(x)\xi_g \quad \text{and} \quad h(x) = \Phi^{(h)}(x)\xi_h. \tag{4.12}$$

To determine the sparse coefficients $\{\xi_k, \xi_g, \xi_h\}$, we can employ the RK4-SINDy framework, and algorithms 1 and 2. We will illustrate this approach to discover enzyme kinetics given by a rational function in §d .

We note that learning a rational dynamical mode with a small denominator may lead to numerical challenges. This could be related to fast transient behaviour, as the gradient can be significantly larger when the denominator is small. Therefore, such cases need to be appropriately handled, for example, with proper data normalization and sampling, although, in our experiment to identify a Michaelis–Menten kinetic model from data (see §5d), we have not noticed any unexpected behaviour.

## (b) Discovering parametric and externally controlled equations

The RK4-SINDy framework immediately embraces the discovery of governing equations that are parametric and externally controlled. Let us begin with an externally controlled dynamical model of the form

$$\dot{x}(t) = f(x(t), u(t)), \tag{4.13}$$

where $x(t) \in \mathbb{R}^n$ and $u(t) \in \mathbb{R}^m$ are state and control input vectors. The goal here is to discover $f(x(t), u(t))$ using the state trajectory $x(t)$ generated using a control input $u(t)$. We aim at

discovering governing equations using dictionary-based identification. As discussed in §2, we construct a symbolic dictionary $\boldsymbol{\Phi}$ of possible candidate features using $\mathbf{x}$ and $\mathbf{u}$, i.e.

$$\boldsymbol{\Phi}(\mathbf{x}, \mathbf{u}) = \left[1, \mathbf{x}, \mathbf{u}, \mathbf{x_u}^{\mathcal{P}_2}, \mathbf{x_u}^{\mathcal{P}_3}\right], \tag{4.14}$$

where $\mathbf{x_u}^{\mathcal{P}_i}$ comprises polynomial terms of degree-$i$, i.e. $\mathbf{x_u}^{\mathcal{P}_2}$ contains degree-2 polynomial terms including cross terms

$$\mathbf{x_u}^{\mathcal{P}_2} = \left[\mathbf{x}_1^2, \dots, \mathbf{x}_n^2, \mathbf{u}_1^2, \dots, \mathbf{u}_m^2, \mathbf{x}_1\mathbf{u}_1, \dots, \mathbf{x}_n\mathbf{u}_1, \mathbf{x}_1\mathbf{u}_2, \dots, \mathbf{x}_n\mathbf{u}_m\right], \tag{4.15}$$

where $\mathbf{u}_i$ is the $i$th element of $\mathbf{u}$. Using measurements of $\mathbf{x}$ and $\mathbf{u}$, we can cast the problem exactly as done in §2 by assuming that $\mathbf{f}(\mathbf{x}(t), \mathbf{u}(t))$ can be determined by selecting appropriate functions from the dictionary $\boldsymbol{\Phi}(\mathbf{x}, \mathbf{u})$. Similarly, system parameters can also be incorporated to discover parametric differential equations of the form

$$\dot{\mathbf{x}}(t; \boldsymbol{\mu}) = \mathbf{f}(\mathbf{x}(t; \boldsymbol{\mu}), \boldsymbol{\mu}), \tag{4.16}$$

where $\boldsymbol{\mu} \in \mathbb{R}^p$ contains the system parameters. It can be considered as a special case of (4.13) since a constant input can be thought of as a parameter in the course of discovering governing equations. We illustrate the RK4-SINDy framework for discovering parametrized Hopf normal form using measurement data (see §5e).

## (c) Parameterized dictionary

The success of the sparse identification highly depends on the quality of the constructed feature dictionary. In other words, the dictionary should contain the right features in which governing differential equations can be given as a linear combination of a few terms from the dictionary. However, it becomes a challenging task when one aims at including, for instance, trigonometric or exponential functions (e.g. $\sin(ax)$, $\mathbf{e}^{(bx)}$), where $\{a, b\}$ are unknown. In an extreme case, one might think of including $\sin(\cdot)$ and $\mathbf{e}^{(\cdot)}$ for each possible value of $a$ and $b$. It would lead to a dictionary of infinite dimensions, hence becoming intractable. To illustrate it, we consider the following governing equation:

$$\dot{\mathbf{x}}(t) = -\mathbf{x}(t) + \exp(-1.75\mathbf{x}(t)). \tag{4.17}$$

Let us assume that we are concerned about discovering the model (4.17) using a time history of $\mathbf{x}(t)$ without any prior knowledge except that we expect exponential nonlinearities based on expert. For instance, an electrical circuit containing diode components typically involves exponential nonlinearities, but the corresponding coefficient is unknown.

We conventionally build a dictionary containing exponential functions using several possible coefficients as follows:

$$\boldsymbol{\Phi}(\mathbf{x}) = \left[1, \mathbf{x}, \mathbf{x}^2, \mathbf{x}^3, \dots, \mathbf{e}^{\mathbf{x}}, \mathbf{e}^{-\mathbf{x}}, \mathbf{e}^{2\mathbf{x}}, \mathbf{e}^{-2\mathbf{x}} \dots, \sin(\mathbf{x}), \cos(\mathbf{x}), \dots\right]. \tag{4.18}$$

However, it is impossible to add infinitely many exponential terms with different coefficients in the dictionary. As a remedy, we discuss the idea of a parameterized dictionary that was also discussed in [44]

$$\boldsymbol{\Phi}_\eta(\mathbf{x}) = \left[1, \mathbf{x}, \mathbf{x}^2, \mathbf{x}^3, \dots, \sin(\eta_1\mathbf{x}), \cos(\eta_2\mathbf{x}), \sin(\eta_3\mathbf{x}^2), \cos(\eta_4\mathbf{x}^2), \dots, \mathbf{e}^{\eta_5\mathbf{x}}, \mathbf{e}^{\eta_6\mathbf{x}^2}, \dots,\right]. \tag{4.19}$$

In this case, we do not need to include all frequencies for trigonometric functions and coefficients for exponential functions. However, it comes at the cost of finding suitable coefficients $\{\eta_i\}$, along with a vector, selecting the right features from the dictionary. Since we solve optimization problems, e.g. (2.10) using gradient descent, we can easily incorporate the parameters $\eta_i$ along with $\xi_i$ as learning parameters and can readily employ algorithms 1 and 2 with a little alteration.

# 5. Results

Here, we demonstrate the success of RK4-SINDy in discovering governing equations using measurement data through a number of examples[1] of different complexity. In the first subsection, we consider simple illustrative examples, namely, linear and nonlinear damped oscillators. Using the linear damped oscillator, we perform a comprehensive study under various conditions, i.e. the robustness of the approach to sparsely sampled and highly corrupted data. We compare the performance of our approach to discover governing equations with [14]; we refer to it as Std-SINDy[2]. In the second example, we study the chaotic Lorenz example and show that RK4-SINDy determines the governing equations, exhibiting the chaotic behaviour accurately. In the third example, we discover neural dynamics from measurement data using RK4-SINDy. As the fourth example, we illustrate the discovery of a model that describes the dynamics of enzymatic activity and contains rational nonlinearities. In the last example, we showcase that RK4-SINDy also successfully discovers the parametric Hopf normal form from collected noisy measurement data for various parameters. Lastly, we mention that we have generated the data using the adaptive solver `solve_ivp` from the python library `SciPy` with default settings. We have implemented algorithms 1 and 2 using the PyTorch library and have used a gradient descent method with a fixed learning rate to solve equation (2.10).

## (a) Two-dimensional damped oscillators

As simple illustrative examples, we consider two-dimensional damped harmonic oscillators. These can be given by linear and nonlinear models. We begin by considering the linear one.

### (i) Linear damped oscillator

Consider a two-dimensional linear damped oscillator whose dynamics is given by

$$\dot{\mathbf{x}}(t) = -0.1\mathbf{x}(t) + 2.0\mathbf{y}(t) \tag{5.1a}$$

and

$$\dot{\mathbf{y}}(t) = -2.0\mathbf{x}(t) - 0.1\mathbf{y}(t). \tag{5.1b}$$

To infer governing equations from measurement data, we first assume to have clean data at a regular time-step $dt$. We then build a symbolic dictionary containing polynomial nonlinearities up to degree 5. Next, we learn governing equations using RK4-SINDy (algorithm 1 with $\lambda = 5 \times 10^{-2}$) and observe the quality of inferred equations for different $dt$. We also present a comparison with Std-SINDy.

The results are shown in figure 2 and table 1. We note that RK4-SINDy is much more robust with the variation in time-step when compared with Std-SINDy, and discovers the governing equations accurately. We also emphasize that for large time-steps, Std-SINDy fails to capture the part of dynamics; in fact, for a time-step $dt = 5 \times 10^{-1}$, Std-SINDy even yields unstable models, figure 2d.

Next, we study the performance of both methodologies under corrupted data. We corrupt the measurement data by adding zero-mean Gaussian white noise of different variances. We present the results in figure 3 and table 2 and note that RK4-SINDy can discover better quality sparse parsimonious models as compared to Std-SINDy even under significantly corrupted data. It is predominantly visible in figure 3d. Naturally, RK4-SINDy also breaks down for a very large amount of noise in measurements, but this breakdown happens much later than for Std-SINDy.

---

[1]Most of all examples are taken from [14].

[2]We use the Python implementation of the method, the so-called `PySINDy` [45].

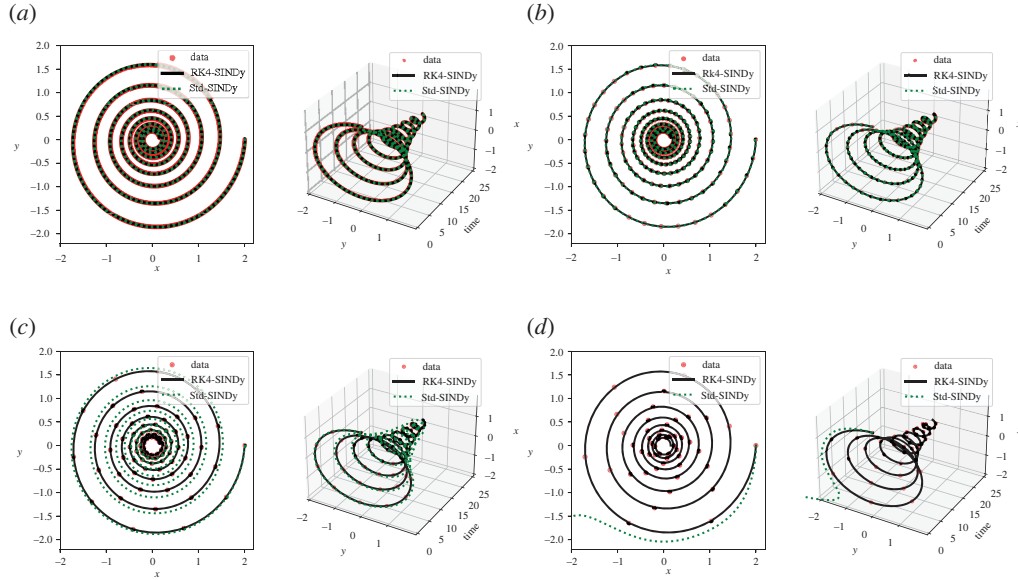

**Figure 2.** Linear two-dimensional model: identified models using data at various regular time-steps. (*a*) Time-step $dt = 1 \times 10^{-2}$, (*b*) time-step $dt = 1 \times 10^{-1}$, (*c*) time-step $dt = 3 \times 10^{-1}$, (*d*) time-step $dt = 5 \times 10^{-1}$. (Online version in colour.)

**Table 1.** Linear two-dimensional model: the discovered governing equations using RK4-SINDy and Std-SINDy are reported for different regular time-steps at which data are collected.

| time-step | RK4-SINDy | Std-SINDy |
|---|---|---|
| $1 \times 10^{-2}$ | $\dot{\mathbf{x}}(t) = -0.100\mathbf{x}(t) + 2.000\mathbf{y}(t)$<br>$\dot{\mathbf{y}}(t) = -2.001\mathbf{x}(t) - 0.100\mathbf{y}(t)$ | $\dot{\mathbf{x}}(t) = -0.100\mathbf{x}(t) + 2.000\mathbf{y}(t)$<br>$\dot{\mathbf{y}}(t) = -2.000\mathbf{x}(t) - 0.100\mathbf{y}(t)$ |
| $1 \times 10^{-1}$ | $\dot{\mathbf{x}}(t) = -0.100\mathbf{x}(t) + 2.001\mathbf{y}(t)$<br>$\dot{\mathbf{y}}(t) = -2.001\mathbf{x}(t) - 0.100\mathbf{y}(t)$ | $\dot{\mathbf{x}}(t) = -0.098\mathbf{x}(t) + 1.987\mathbf{y}(t)$<br>$\dot{\mathbf{y}}(t) = -1.988\mathbf{x}(t) - 0.098\mathbf{y}(t)$ |
| $3 \times 10^{-1}$ | $\dot{\mathbf{x}}(t) = -0.101\mathbf{x}(t) + 2.002\mathbf{y}(t)$<br>$\dot{\mathbf{y}}(t) = -2.002\mathbf{x}(t) - 0.101\mathbf{y}(t)$ | $\dot{\mathbf{x}}(t) = -0.078\mathbf{x}(t) + 1.884\mathbf{y}(t)$<br>$\dot{\mathbf{y}}(t) = -1.906\mathbf{x}(t) - 0.084\mathbf{y}(t)$ |
| $5 \times 10^{-1}$ | $\dot{\mathbf{x}}(t) = -0.103\mathbf{x}(t) + 2.011\mathbf{y}(t)$<br>$\dot{\mathbf{y}}(t) = -2.011\mathbf{x}(t) - 0.103\mathbf{y}(t)$ | $\dot{\mathbf{x}}(t) = 1.688\mathbf{y}(t)$<br>$\dot{\mathbf{y}}(t) = -1.864\mathbf{x}(t) - 0.123\mathbf{x}(t)^2$<br>$\quad - 0.146\mathbf{x}(t)^2\mathbf{y}(t) + 0.115\mathbf{x}(t)^4\mathbf{y}(t)$<br>$\quad + 0.133\mathbf{x}(t)^3\mathbf{y}(t)$ |
| $7 \times 10^{-1}$ | $\dot{\mathbf{x}}(t) = -0.121\mathbf{x}(t) + 1.964\mathbf{y}(t)$<br>$\quad + 0.153\mathbf{x}^2(t)\mathbf{y}(t) + 0.068\mathbf{x}(t)\mathbf{y}^2(t)$<br>$\quad - 0.081\mathbf{x}^4(t)\mathbf{y}(t) - 0.063\mathbf{x}^3(t)\mathbf{y}^2(t)$<br>$\dot{\mathbf{y}}(t) = -2.024\mathbf{x}(t) + -0.078\mathbf{y}(t)$ | $\dot{\mathbf{x}}(t) = 1.415\mathbf{y}(t)$<br>$\dot{\mathbf{y}}(t) = -1.385\mathbf{x}(t) + 0.121\mathbf{x}(t)^2$<br>$\quad - 0.116\mathbf{x}(t)^4 - 0.057\mathbf{x}(t)^5$<br>$\quad + 0.074\mathbf{x}(t)^3\mathbf{y}(t)^2$ |

### (ii) Cubic damped oscillator

Next, we consider a cubic damped oscillator, governed by

$$\left.\begin{array}{l}\dot{\mathbf{x}}(t) = -0.1\mathbf{x}(t)^3 + 2.0\mathbf{y}(t)^3 \\[4pt] \dot{\mathbf{y}}(t) = -2.0\mathbf{x}(t)^3 - 0.1\mathbf{y}(t)^3.\end{array}\right\} \tag{5.2}$$

and

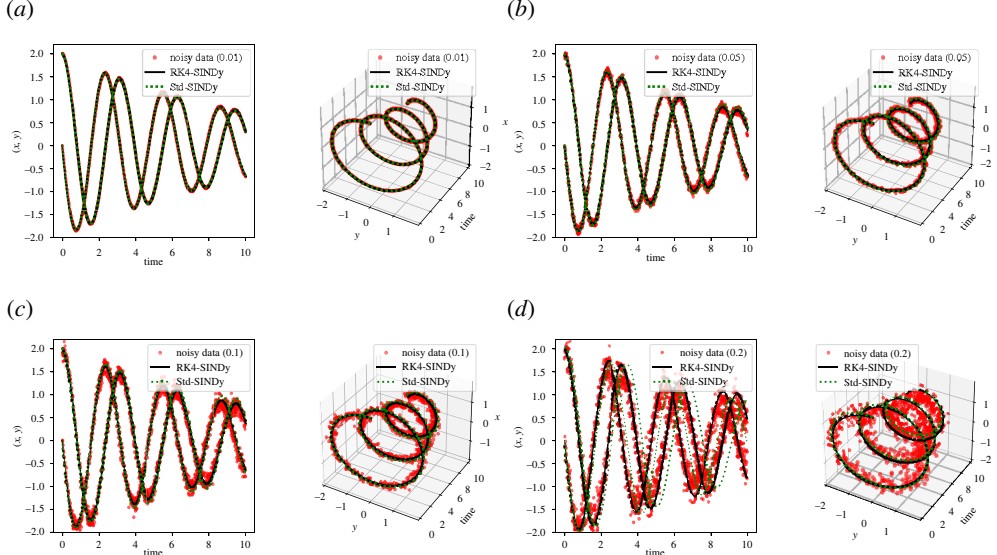

**Figure 3.** Linear two-dimensional model: the transient responses of discovered models using corrupted data are compared. (*a*) Noise level $\sigma = 1 \times 10^{-2}$, (*b*) noise level $\sigma = 5 \times 10^{-2}$, (*c*) noise level $\sigma = 1 \times 10^{-1}$, (*d*) noise level $\sigma = 2 \times 10^{-1}$. (Online version in colour.)

**Table 2.** Linear two-dimensional model: the discovered governing equations, by employing RK4-SINDy and Std-SINDy, are reported. In this scenario, the measurement data are corrupted using zero-mean Gaussian white noise of different variances.

| noise level (SNR) | RK4-SINDy | Std-SINDy |
|---|---|---|
| $1 \times 10^{-2}$ (39.46 dB) | $\dot{\mathbf{x}}(t) = -0.099\mathbf{x}(t) + 1.999\mathbf{y}(t)$ $\dot{\mathbf{y}}(t) = -2.000\mathbf{x}(t) - 0.101\mathbf{y}(t)$ | $\dot{\mathbf{x}}(t) = -0.102\mathbf{x}(t) + 1.999\mathbf{y}(t)$ $\dot{\mathbf{y}}(t) = -2.002\mathbf{x}(t) - 0.101\mathbf{y}(t)$ |
| $5 \times 10^{-2}$ (25.35 dB) | $\dot{\mathbf{x}}(t) = -0.095\mathbf{x}(t) + 1.999\mathbf{y}(t)$ $\dot{\mathbf{y}}(t) = -1.995\mathbf{x}(t) - 0.105\mathbf{y}(t)$ | $\dot{\mathbf{x}}(t) = -0.078\mathbf{x}(t) + 2.001\mathbf{y}(t)$ $\dot{\mathbf{y}}(t) = -1.995\mathbf{x}(t) - 0.105\mathbf{y}(t)$ |
| $1 \times 10^{-1}$ (19.36 dB) | $\dot{\mathbf{x}}(t) = -0.091\mathbf{x}(t) + 1.985\mathbf{y}(t)$ $\dot{\mathbf{y}}(t) = -1.997\mathbf{x}(t) - 0.103\mathbf{y}(t)$ | $\dot{\mathbf{x}}(t) = -0.076\mathbf{x}(t) + 1.969\mathbf{y}(t)$ $\dot{\mathbf{y}}(t) = -2.008\mathbf{x}(t) - 0.095\mathbf{y}(t)$ |
| $2 \times 10^{-1}$ (13.06 dB) | $\dot{\mathbf{x}}(t) = -0.177\mathbf{x}(t) + 2.053\mathbf{y}(t)$ $- 0.063\mathbf{x}^2\mathbf{y} + 0.059\mathbf{xy}^2$ $\dot{\mathbf{y}}(t) = -1.960\mathbf{x}(t)$ | $\dot{\mathbf{x}}(t) = -0.173\mathbf{x}(t) + 1.950\mathbf{y}(t) - 0.056\mathbf{y}(t)^2$ $+ 0.059\mathbf{x}(t)^3 - 0.079\mathbf{x}(t)^2\mathbf{y} + 0.095$ $\dot{\mathbf{y}}(t) = -2.005\mathbf{x}(t) - 0.095\mathbf{y}(t) + 0.069\mathbf{x}(t)\mathbf{y}(t)$ $+ 0.062\mathbf{x}(t)^3 + 0.060\mathbf{x}(t)\mathbf{y}(t)^2$ |
| $3 \times 10^{-1}$ (9.93 dB) | $\dot{\mathbf{x}}(t) = 2.167\mathbf{y} - 0.079\mathbf{xy}$ $- 0.101\mathbf{x}^2\mathbf{y} - 0.119\mathbf{xy}^2$ $- 0.085\mathbf{y}^3$ $\dot{\mathbf{y}}(t) = -2.026\mathbf{x} - 0.135\mathbf{y}$ $- 0.066\mathbf{x}^2\mathbf{y} + 0.056\mathbf{y}^3$ | $\dot{\mathbf{x}}(t) = 0.113 + 2.271\mathbf{y} - 0.165\mathbf{x}^2$ $- 0.096\mathbf{xy} - 0.062\mathbf{y}^2 - 0.095\mathbf{x}^3$ $- 0.260\mathbf{x}^2\mathbf{y} + -0.159\mathbf{y}^3$ $\dot{\mathbf{y}}(t) = -2.183\mathbf{x} - 0.197\mathbf{y} + 0.083\mathbf{xy}$ $+ 0.131\mathbf{x}^3 + 0.150\mathbf{xy}^2 + 0.057\mathbf{y}^3$ |

Like in the linear case, we aim at discovering the governing equations using measurement data. We repeat the study done in the previous example using different regular time-steps. We report the quality of discovered models using RK4-SINDy and Std-SINDy in figure 4 and table 3. We observe that RK4-SINDy successfully discovers the governing equations quite accurately, whereas Std-SINDy struggles to identify the governing equations when measurements data are collected at a larger time-step. It simply fails to obtain a stable model for the time-step d$t = 0.1$. It showcases the robustness of the RK4-SINDy to discover interpretable models even when data are collected sparsely.

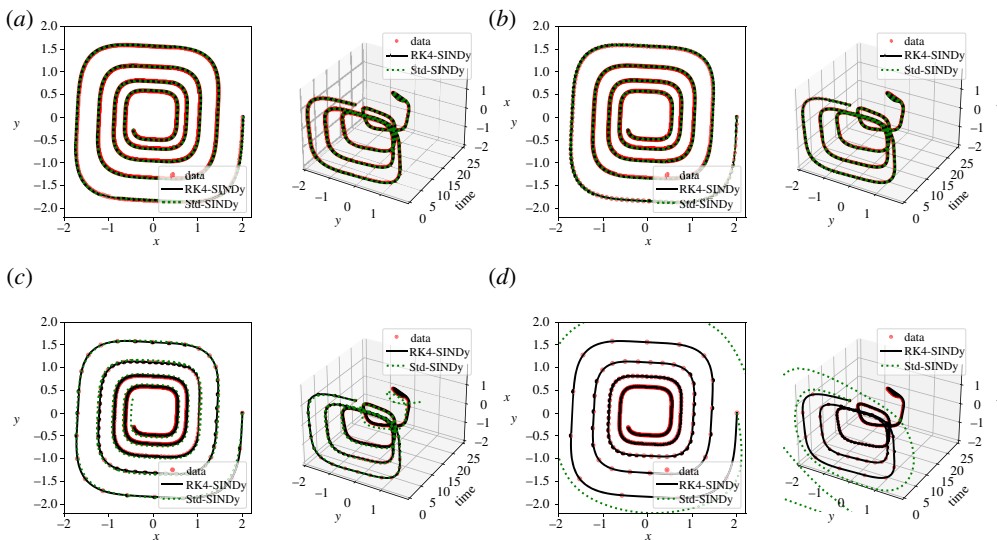

**Figure 4.** Cubic two-dimensional model: a comparison of the transient responses of discovered models using data at different regular time-steps. (*a*) time-step d$t = 5 \times 10^{-3}$, (*b*) time-step d$t = 1 \times 10^{-2}$, (*c*) time-step d$t = 5 \times 10^{-2}$ and (*d*) time-step d$t = 1 \times 10^{-1}$. (Online version in colour.)

**Table 3.** Cubic two-dimensional model: the table reports the discovered governing equations by employing RK4-SINDy and Std-SINDy.

| time-step | RK4-SINDy | Std-SINDy |
|---|---|---|
| $5 \times 10^{-3}$ | $\dot{\mathbf{x}}(t) = -0.099\mathbf{x}(t)^3 + 1.996\mathbf{y}(t)^3$ <br> $\dot{\mathbf{y}}(t) = -1.997\mathbf{x}(t)^3 - 0.100\mathbf{y}(t)^3$ | $\dot{\mathbf{x}}(t) = -0.099\mathbf{x}(t)^3 + 1.995\mathbf{y}(t)^3$ <br> $\dot{\mathbf{y}}(t) = -1.996\mathbf{x}(t)^3 - 0.099\mathbf{y}(t)^3$ |
| $1 \times 10^{-2}$ | $\dot{\mathbf{x}}(t) = -0.099\mathbf{x}(t)^3 + 1.995\mathbf{y}(t)^3$ <br> $\dot{\mathbf{y}}(t) = -1.997\mathbf{x}(t)^3 - 0.100\mathbf{y}(t)^3$ | $\dot{\mathbf{x}}(t) = -0.100\mathbf{x}(t)^3 + 1.994\mathbf{y}(t)^3$ <br> $\dot{\mathbf{y}}(t) = -1.996\mathbf{x}(t)^3 - 0.099\mathbf{y}(t)^3$ |
| $5 \times 10^{-2}$ | $\dot{\mathbf{x}}(t) = -0.100\mathbf{x}(t)^3 + 1.995\mathbf{y}(t)^3$ <br> $\dot{\mathbf{y}}(t) = -1.997\mathbf{x}(t)^3 - 0.100\mathbf{y}(t)^3$ | $\dot{\mathbf{x}}(t) = -0.092\mathbf{x}(t)^3 + 2.002\mathbf{y}(t)^3$ <br> $+ 0.076\mathbf{x}^4\mathbf{y} - 0.107\mathbf{x}^2\mathbf{y}^3$ <br> $\dot{\mathbf{y}}(t) = -1.981\mathbf{x}(t)^3 - 0.092\mathbf{y}(t)^3$ <br> $+ 0.078\mathbf{x}^3\mathbf{y}^2 - 0.068\mathbf{x}\mathbf{y}^4$ |
| $1 \times 10^{-1}$ | $\dot{\mathbf{x}}(t) = -0.103\mathbf{x}(t)^3 + 2.000\mathbf{y}(t)^3$ <br> $\dot{\mathbf{y}}(t) = -2.001\mathbf{x}(t)^3 - 0.098\mathbf{y}(t)^3$ | $\dot{\mathbf{x}}(t) = 0.090\mathbf{x}(t) - 0.097\mathbf{x}(t)^2 - 0.463\mathbf{x}(t)^3$ <br> $+ \cdots + 0.381\mathbf{x}(t)^3\mathbf{y}(t)^2 - 0.258\mathbf{x}(t)\mathbf{y}(t)^4$ <br> $\dot{\mathbf{y}}(t) = 0.100\mathbf{x}(t) + 0.104\mathbf{x}(t)^2 + 0.051\mathbf{x}(t)\mathbf{y}(t)$ <br> $+ \cdots + 0.381\mathbf{x}(t)^3\mathbf{y}(t)^2 - 0.258\mathbf{x}(t)\mathbf{y}(t)^4$ |

## (b) Fitz–Hugh Nagumo model

Here, we explore the discovery of the nonlinear FHN model that describes the activation and deactivation of neurons in a simplistic way [46]. The governing equations are

$$\left.\begin{aligned} \dot{\mathbf{v}}(t) &= \mathbf{v}(t) - \mathbf{w}(t) - \frac{1}{3}\mathbf{v}(t)^3 + 0.5 \\ \dot{\mathbf{w}}(t) &= 0.040\mathbf{v}(t) - 0.028\mathbf{w}(t) + 0.032. \end{aligned}\right\} \tag{5.3}$$

and

We collect the time-history data of $\mathbf{v}(t)$ and $\mathbf{w}(t)$ using homogeneous initial conditions. We construct a dictionary containing polynomial terms up to the third degree. We employ

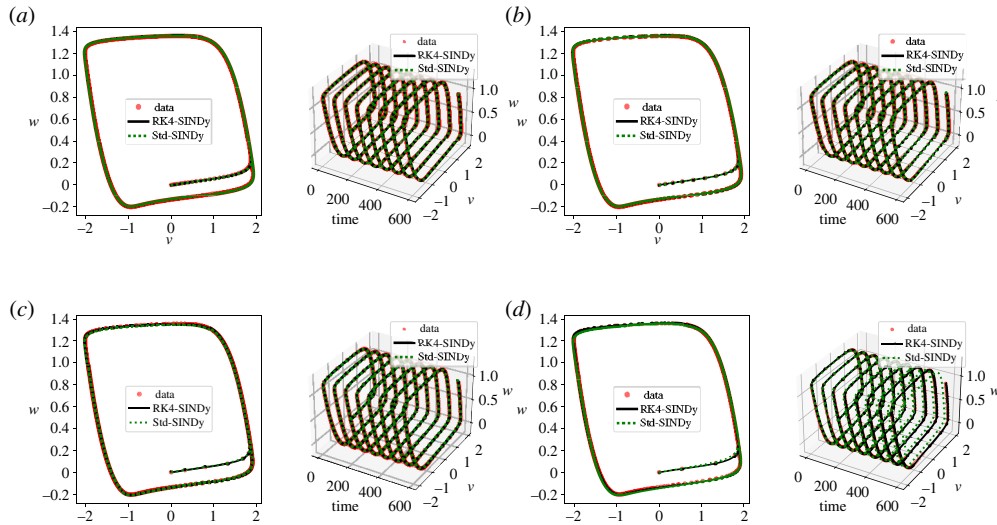

**Figure 5.** FHN model: a comparison of the transient responses of the discovered differential equations using data collected at different regular time-steps. (a) time-step $dt = 1.0 \times 10^{-1}$, (b) time-step $dt = 2.5 \times 10^{-1}$, (c) time-step $dt = 5.0 \times 10^{-1}$ and (d) time-step $dt = 7.5 \times 10^{-1}$. (Online version in colour.)

**Table 4.** FHN model: discovered models using data at various time-steps using RK4-SINDy and Std-SINDy.

| dt | RK4-SINDy | Std-SINDy |
|---|---|---|
| $1.0 \times 10^{-1}$ | $\dot{\mathbf{v}}(t) = 0.499 + 0.998\mathbf{v} - 0.998\mathbf{w} - 0.333\mathbf{v}^3$ <br> $\dot{\mathbf{w}}(t) = 0.032 + 0.040\mathbf{v} - 0.028\mathbf{w}$ | $\dot{\mathbf{v}}(t) = 0.498 + 0.996\mathbf{v} - 0.996\mathbf{w} - 0.332\mathbf{v}^3$ <br> $\dot{\mathbf{w}}(t) = 0.032 + 0.040\mathbf{v} - 0.028\mathbf{w}$ |
| $2.5 \times 10^{-1}$ | $\dot{\mathbf{v}}(t) = 0.499 + 0.998\mathbf{v} - 0.998\mathbf{w} - 0.333\mathbf{v}^3$ <br> $\dot{\mathbf{w}}(t) = 0.032 + 0.040\mathbf{v} - 0.028\mathbf{w}$ | $\dot{\mathbf{v}}(t) = 0.494 + 0.985\mathbf{v} - 0.989\mathbf{w} - 0.328\mathbf{v}^3$ <br> $\dot{\mathbf{w}}(t) = 0.032 + 0.040\mathbf{v} - 0.028\mathbf{w}$ |
| $5.0 \times 10^{-1}$ | $\dot{\mathbf{v}}(t) = 0.501 + 1.001\mathbf{v} - 1.001\mathbf{w} - 0.334\mathbf{v}^3$ <br> $\dot{\mathbf{w}}(t) = 0.032 + 0.040\mathbf{v} - 0.028\mathbf{w}$ | $\dot{\mathbf{v}}(t) = 0.482 + 0.943\mathbf{v} - 0.959\mathbf{w}$ <br> $\quad - 0.034\mathbf{vw} - 0.311\mathbf{v}^3 + 0.024\mathbf{vw}^2$ <br> $\dot{\mathbf{w}}(t) = 0.032 + 0.040\mathbf{v} - 0.028\mathbf{w}$ |
| $7.5 \times 10^{-1}$ | $\dot{\mathbf{v}}(t) = 0.502 + 1.001\mathbf{v} - 1.003\mathbf{w} - 0.334\mathbf{v}^3$ <br> $\dot{\mathbf{w}}(t) = 0.032 + 0.040\mathbf{v} - 0.027\mathbf{w}$ | $\dot{\mathbf{v}}(t) = 0.459 + 0.816\mathbf{v} - 0.982\mathbf{w}$ <br> $\quad - 0.013\mathbf{v}^2 + \cdots + 0.131\mathbf{vw}^2 - 0.021\mathbf{w}^3$ <br> $\dot{\mathbf{w}}(t) = 0.032 + 0.040\mathbf{v} - 0.028\mathbf{w}$ |

RK4-SINDy (algorithm 1 with $\lambda = 10^{-2}$) and Std-SINDy. We discover governing equations using the data collected in the time interval $[0, 600]$s. We identify models under different conditions, namely, different time-steps at which data are collected. We report the results in figure 5 and table 4. It can be observed that RK4-SINDy faithfully discovers the underlying governing equations by picking the correct features from the dictionary and estimates the corresponding coefficients up to 1% accurately. On the other hand, Std-SINDy breaks down when data are taken at a large time-step.

## (c) Chaotic Lorenz system

As the next example, we consider the problem of discovering the nonlinear Lorenz model [47]. The dynamics of the chaotic system is governed by

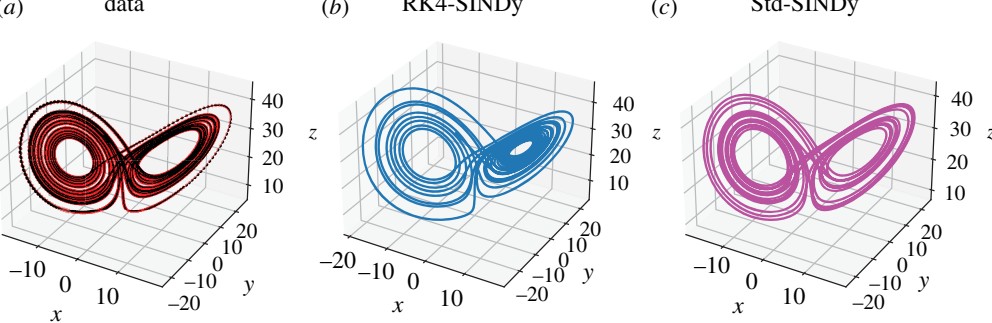

**Figure 6.** Chaotic Lorenz model: (*a*) the collected data (in dotted) and a finely spaced trajectory of the ground truth is shown. (*b,c*) The trajectories obtained using the discovered models using RK4-SINDy and Std-SINDy, respectively. (Online version in colour.)

**Table 5.** Chaotic Lorenz model: discovered governing equations using RK4-SINDy and Std-SINDy.

| RK4-SINDy | | Std-SINDy |
|---|---|---|
| $\dot{\tilde{\mathbf{x}}}(t) = -10.004\tilde{\mathbf{x}}(t) + 10.004\tilde{\mathbf{y}}(t),$ | | $\dot{\tilde{\mathbf{x}}}(t) = -9.983\tilde{\mathbf{x}}(t) + 9.983\tilde{\mathbf{y}}(t),$ |
| $\dot{\tilde{\mathbf{y}}}(t) = 2.966\tilde{\mathbf{x}} - 0.956\tilde{\mathbf{y}}(t) - 7.953\tilde{\mathbf{x}}(t)\tilde{\mathbf{z}}(t),$ | | $\dot{\tilde{\mathbf{y}}}(t) = 2.912\tilde{\mathbf{x}} - 0.922\tilde{\mathbf{y}}(t) - 7.911\tilde{\mathbf{x}}(t)\tilde{\mathbf{z}}(t),$ |
| $\dot{\tilde{\mathbf{z}}}(t) = 7.944\tilde{\mathbf{x}}(t)\tilde{\mathbf{y}}(t) - 2.669\tilde{\mathbf{z}}(t) - 8.336$ | | $\dot{\tilde{\mathbf{z}}}(t) = 7.972\tilde{\mathbf{x}}(t)\tilde{\mathbf{y}}(t) - 2.662\tilde{\mathbf{z}}(t) - 8.313$ |

$$\left.\begin{aligned}\dot{\mathbf{x}}(t) &= -10\mathbf{x}(t) + 10\mathbf{y}(t),\\ \dot{\mathbf{y}}(t) &= \mathbf{x}(28 - \mathbf{z}(t)) - \mathbf{y}(t)\\ \dot{\mathbf{z}}(t) &= \mathbf{x}(t)\mathbf{y}(t) - \tfrac{8}{3}\mathbf{z}(t).\end{aligned}\right\} \tag{5.4}$$

and

We collect the data by simulating the model from time $t = 0$ to $t = 20$ with a time-step of $dt = 10^{-2}$. To discover the governing equations using the measurement data, we employ RK4-SINDy and Std-SINDy with the fixed cut-off parameter $\lambda = 0.5$. However, before employing the methodologies, we perform a normalization step. The reason behind this is that the mean value of the variable $\mathbf{z}$ is large, and the standard deviations of all three variables are much larger than 1. Consequently, a dictionary containing polynomial terms would be highly ill-conditioned. To circumvent this, we perform a normalization of data. Ideally, one performs normalization such that the mean and variance of the transformed data are 0 and 1. But for this particular example, we normalize such that the interactions between the transformed variables are similar to (5.4). Hence, we propose a transformation as

$$\tilde{\mathbf{x}}(t) := \tfrac{\mathbf{x}(t)}{8}, \quad \tilde{\mathbf{y}}(t) := \tfrac{\mathbf{y}(t)}{8} \quad \text{and} \quad \tilde{\mathbf{z}}(t) := \tfrac{\mathbf{z}(t) - 25}{8}. \tag{5.5}$$

Consequently, we obtain a model

$$\left.\begin{aligned}\dot{\tilde{\mathbf{x}}}(t) &= -10\tilde{\mathbf{x}}(t) + 10\tilde{\mathbf{y}}(t),\\ \dot{\tilde{\mathbf{y}}}(t) &= \tilde{\mathbf{x}}(28 - 8\tilde{\mathbf{z}}(t)) - \tilde{\mathbf{y}}(t)\\ \dot{\tilde{\mathbf{z}}}(t) &= 8\tilde{\mathbf{x}}(t)\tilde{\mathbf{y}}(t) - \tfrac{8}{3}\tilde{\mathbf{z}}(t) - \tfrac{25}{3}.\end{aligned}\right\} \tag{5.6}$$

and

The models (5.4) and (5.6) look alike, and the basis features in which the dynamics of both models lie are the same except for a constant. However, an appealing property of the model (5.6) or the transformed data is that the data becomes well-conditioned, hence the dictionary containing polynomial features. Next, we discover models by employing RK4-SINDy and Std-SINDy using the transformed data. For this, we construct a dictionary with polynomial nonlinearities up to

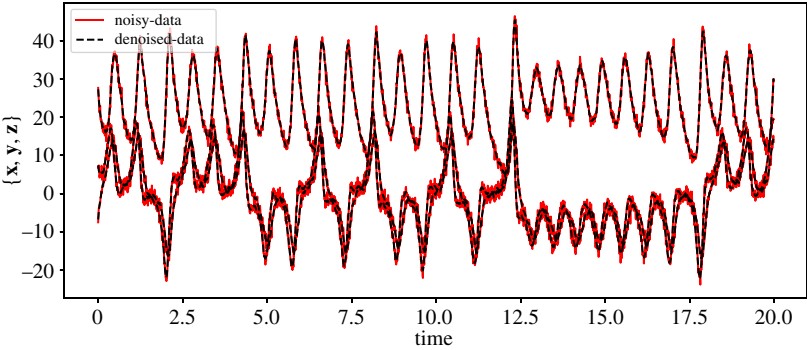

**Figure 7.** Chaotic Lorenz model: the figure shows the noisy measurements of $\{\mathbf{x}, \mathbf{y}, \mathbf{z}\}$ that are corrupted by adding zero-mean Gaussian noise of variance one. It also shows the denoised signal done using a Savitzky–Golay filter [48]. (Online version in colour.)

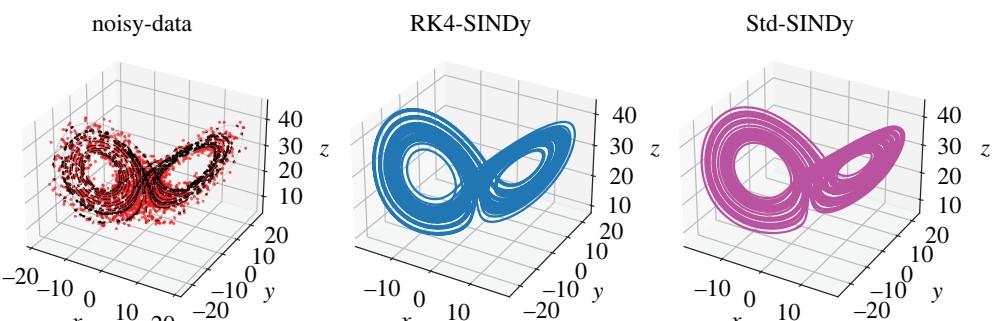

**Figure 8.** Chaotic Lorenz model: the left figure shows the collected noisy data (in dotted), and a continuous trajectory of the ground truth is shown. We have added Gaussian white noise of mean zero and variance one. The middle and right figures present the transient trajectories obtained using the discovered models using RK4-SINDy and Std-SINDy, respectively, and show that the dynamics of the discovered models are intact on the attractor. (Online version in colour.)

degree 3. We observe the result in figure 6 and table 5. We note that both methods identify correct features from the dictionary with coefficients that are close to the ground truth, but the RK4-SINDy model coefficients are closer to the ground-truth ones. It is also worthwhile to note that the coefficients of the obtained RK4-SINDy model are only 0.01% off the ground-truth, but the dynamics still seem quite different, figure 6. A reason behind this is the highly chaotic behaviour of the dynamics. As a result, a tiny deviation in the coefficients can significantly impact the transient behaviour in an absolute sense; however, the dynamics on the attractor are perfectly captured. Next, we study the performance of the approaches under noisy measurements. For this, we add zero-mean Gaussian white noise of variance one. To employ RK4-SINDy, we first apply a Savitzky–Golay filter [48] to denoise the time-history data, figure 7. For Std-SINDy as well, we use the same filter to denoise the signal and approximate the derivative information. We plot the trajectories of the discovered models and ground-truth in figure 8 and observe that dynamics on the attractor is still intact; however, we note that the discovered equations are very different from the ground truth, table 6. The learning can be improved by employing algorithm 2, where we iteratively remove the smallest coefficient and determine the sparsest solutions by looking at the Pareto-front. However, it comes at a slightly higher computational cost.

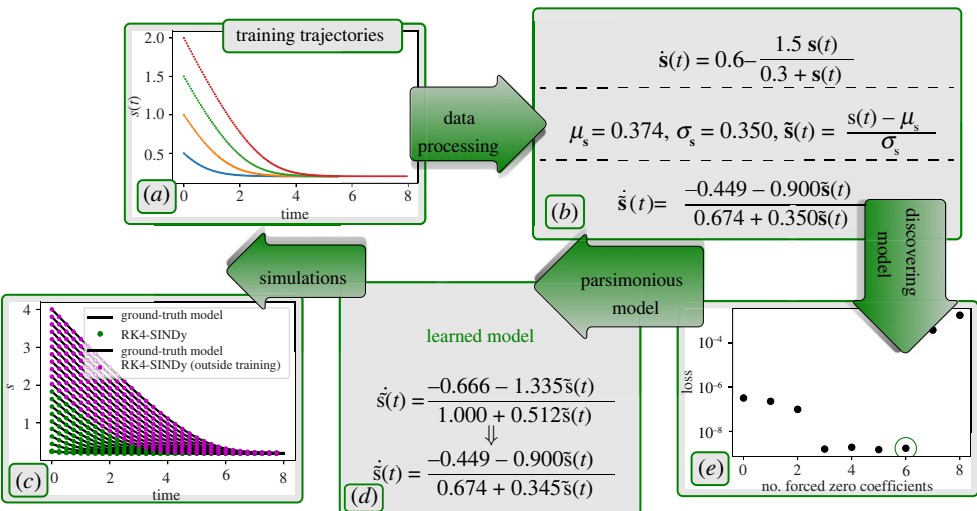

**Figure 9.** Michaelis–Menten kinetics: in the first step, we have collected data for four initial conditions at a time-stepping $dt = 5 \times 10^{-2}$. In the second step, we performed data-processing to normalize the data using the mean and standard deviation. In the third step, we employed RK4-SINDy (algorithm 2) to discover the most parsimonious model. For this, we observe the Pareto front and pick the model that best fits the data, yet has the maximum number of zero coefficients. We then compare the discovered model with the ground truth and find that the proposed methodology could find precise candidates from the polynomial dictionary. The corresponding coefficients have less than 1% errors. (Online version in colour.)

**Table 6.** Lorenz model: discovered governing equations using RK4-SINDy and Std-SINDy from noisy measurements.

| RK4-SINDy | Std-SINDy |
|---|---|
| $\dot{\tilde{\mathbf{x}}}(t) = -9.016\tilde{\mathbf{x}} + 8.221\tilde{\mathbf{y}} - 1.675\tilde{\mathbf{x}}\tilde{\mathbf{z}}$ | $\dot{\tilde{\mathbf{x}}}(t) = -8.842\tilde{\mathbf{x}} + 8.373\tilde{\mathbf{y}} - 3.107\tilde{\mathbf{x}}\tilde{\mathbf{z}}$ |
| $+ 0.895\tilde{\mathbf{x}}^3 + 0.603\tilde{\mathbf{x}}^2\tilde{\mathbf{y}} + -0.579\tilde{\mathbf{x}}\tilde{\mathbf{y}}^2$ | $+ 1.065\tilde{\mathbf{y}}\tilde{\mathbf{z}} + 2.165\tilde{\mathbf{x}}^3 + -1.215\tilde{\mathbf{x}}^2\tilde{\mathbf{y}}$ |
| $\dot{\tilde{\mathbf{y}}}(t) = 1.025\tilde{\mathbf{y}} - 6.133\tilde{\mathbf{x}}\tilde{\mathbf{z}} - 1.033\tilde{\mathbf{y}}\tilde{\mathbf{z}}$ | $\dot{\tilde{\mathbf{y}}}(t) = 1.811\tilde{\mathbf{x}} + -7.580\tilde{\mathbf{x}}\tilde{\mathbf{z}}$ |
| $\dot{\tilde{\mathbf{z}}}(t) = -8.3451 - 2.708\tilde{\mathbf{z}} + 7.971\tilde{\mathbf{x}}\tilde{\mathbf{y}}$ | $\dot{\tilde{\mathbf{z}}}(t) = -8.3441 + -2.710\tilde{\mathbf{z}} + 7.950\tilde{\mathbf{x}}\tilde{\mathbf{y}}$ |

## (d) Michaelis–Menten kinetics

To illustrate RK4-SINDy to discover governing equations that are given by a fraction of two nonlinear functions, we consider arguably the most well-known model for enzyme kinetics, namely the Michaelis–Menten model [49,50]. The model explains the dynamics of binding and unbinding of enzymes with a substrate **s**. In a simplistic way, the dynamics are governed in [51]

$$\dot{\mathbf{s}}(t) = 0.6 - \frac{1.5\mathbf{s}(t)}{0.3 + \mathbf{s}(t)}. \tag{5.7}$$

As a first step, we generate data using four initial conditions $\{0.5, 1.0, 1.5, 2.0\}$. We collect data at a time-step $dt = 5 \times 10^{-2}$, figure 9a. Typically, governing equations explaining biological processes involve rational functions. Therefore, we aim at discovering the enzyme kinetics model by assuming a rational form as shown in (4.1), i.e. the vector field of $\mathbf{s}(t)$ takes the form $\mathbf{g}(\mathbf{s}(t))/(1 + \mathbf{h}(\mathbf{s}(t)))$.

Next, in order to identify $\mathbf{g}(\mathbf{s})$ and $\mathbf{h}(\mathbf{s})$, we construct polynomial dictionaries, containing terms up to degree 4. After that, we employ RK4-SINDy to identify the precise features from the dictionaries to characterize $\mathbf{g}$ and $\mathbf{h}$. Moreover, we apply the iterative thresholding approach discussed in algorithm 2, in contrast to previously considered examples where fixed thresholding

is applied. Note that the success of the RK4-SINDy approach not only depends on a dictionary containing candidate features but the quality of data. We note that the dictionary data matrix conditioning improves when data are normalized to mean-zero and variance-one. This is crucial for the polynomial basis in the dictionary. For this example, we normalize the data before employing RK4-SINDy. That is, we apply the following transformation:

$$\tilde{\mathbf{s}}(t) = \frac{\tilde{\mathbf{s}} - \mu_{\mathbf{s}}}{\sigma_{\mathbf{s}}}, \tag{5.8}$$

where $\mu_{\mathbf{s}}$ and $\sigma_{\mathbf{s}}$ are the mean and standard deviation of the collected data. Next, using the normalized data, we learn the governing equation, describing the dynamics of $\tilde{\mathbf{s}}(t)$. Since we consider dictionaries for $\mathbf{g}$ and $\mathbf{h}$, containing polynomials of degree 4, there are in total 9 coefficients. To identify the correct model while employing algorithm 2, we keep track of the loss (data-fidelity) and the number of non-zero coefficients, which is shown in figure 9c. This allows us to build a Pareto front for the optimization problem and to choose the most parsimonious model that describes the dynamics present in the collected data. One of the most attractive features of learning parsimonious models is to avoid over-fitting and generalizing better in regions in which data are not collected. It is exactly what we observed as well. As shown in figure 9e, the learned model predicts dynamics very accurately in the region far away from the training one.

Next, we study the performance of the method under noisy measurements. For this, we corrupt the collected data using zero-mean Gaussian noise of variance $\sigma = 2 \times 10^{-2}$. Then, we process the data by first employing a noise-reduction filter, namely Savitzky–Golay, followed by normalizing the data. In the third step, we focus on learning the most parsimonious model by picking appropriate candidates from the polynomial dictionary. Remarkably, the method allows us to find a model with correct features from the dictionary and coefficient accuracy up to 5%. Furthermore, the model faithfully generalizes to regimes outside the training, even using noisy measurements (figure 10).

## (e) Hopf normal form

In our last example, we study the discovery of parameterized differential equations from noisy measurements. Many real-world dynamical processes have system parameters, and depending on them, the system may exhibit very distinctive dynamical behaviours. To illustrate the efficiency of RK4-SINDy to discover parametric equations, we consider the Hopf system

$$\left. \begin{aligned} \dot{\mathbf{x}}(t) &= \mu\mathbf{x}(t) - \omega\mathbf{y}(t) - \mathbf{A}\mathbf{x}(t)(\mathbf{x}(t)^2 + \mathbf{y}(t)^2) \\ \dot{\mathbf{y}}(t) &= \omega\mathbf{x}(t) + \mu\mathbf{y}(t) - \mathbf{A}\mathbf{y}(t)(\mathbf{x}(t)^2 + \mathbf{y}(t)^2) \end{aligned} \right\} \tag{5.9}$$

and

that exhibits bifurcation with respect to the parameter $\mu$. For this example, we collect measurements for eight different parameter values $\mu$ at a time-step 0.2 by fixing $\omega = 1$ and $\mathbf{A} = 1$. Then, we corrupt the measurement data by adding Gaussian sensor noise of 1% that is shown in figure 11a. Next, we aim at constructing a symbolic polynomial dictionary $\Phi$ by including the parameter $\mu$ as the dependent variables. While building a polynomial dictionary, it is important to choose the degree of the polynomial as well. Moreover, it is known that the polynomial basis becomes numerically unstable as the degree increases. Hence, solving the optimization problem (2.9) becomes challenging. By means of this example, we discuss an assessment test to choose the appropriate degree of the polynomial in the dictionary. Essentially, we inspect data fidelity with respect to the degree of the polynomial in the dictionary. When the dictionary contains all essential polynomial features, then a sharp drop in the error is expected. We observe in figure 11b a sharp drop in the error at the degree 3, and the error remains almost the same even when higher polynomial features are added. It indicates that polynomial degree 3 is sufficient to describe the dynamics. Using the dictionary containing degree 3 polynomial features, we seek to identify the minimum number of features from the dictionary that explains the underlying dynamics. We achieve this by employing RK4-SINDy, and comparing the performance with Std-SINDy.

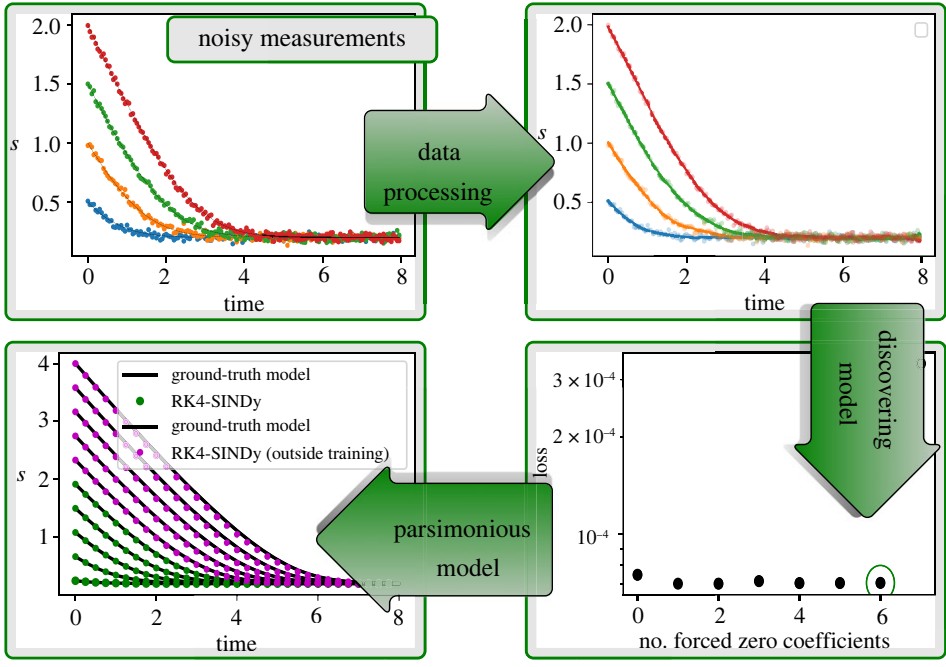

**Figure 10.** Michaelis–Menten kinetics: the figure demonstrates the necessary steps to uncover the most parsimonious model using noisy measurement data. It also testifies to the capability of discovering the most parsimonious model to even generalize beyond the training regime. (Online version in colour.)

**Table 7.** Hopf normal form: here, we report discovered governing equations using noisy measurement data, representing dynamics of Hopf bifurcation. We note that RK4-SINDy recovers the Hopf normal form very accurately; on the other hand, Std-SINDy breaks down.

| method | discovered model |
|---|---|
| RK4-SINDy | $\dot{\mathbf{x}}(t) = 1.001\mu\mathbf{x}(t) - 1.001\mathbf{y}(t) - 0.996\mathbf{x}(t)(\mathbf{x}(t)^2 + \mathbf{y}(t)^2)$ |
| | $\dot{\mathbf{y}}(t) = 1.001\mathbf{x}(t) + 1.010\mu\mathbf{y}(t) - 1.006\mathbf{x}(t)^2\mathbf{y}(t) - 1.004\mathbf{y}(t)^3$ |
| Std-SINDy | $\dot{\mathbf{x}}(t) = -0.961\mathbf{y}(t) + 0.719\mu\mathbf{x}(t) + 0.822\mu\mathbf{y}(t) - 0.735\mathbf{x}(t)^3 - 1.044\mathbf{x}(t)^2\mathbf{y}$ |
| | $\qquad - 0.686\mathbf{x}(t)\mathbf{y}(t)^2 - 0.846\mathbf{y}(t)^3$ |
| | $\dot{\mathbf{y}}(t) = 0.986\mathbf{x}(t) + 0.899\mu\mathbf{y}(t) - 0.882\mathbf{x}(t)^2\mathbf{y}(t) - 0.904\mathbf{y}(t)^3.$ |

We note down the discovered governing equations in table 7, where we note an impressive performance of RK4-SINDy to discover the exact form of the underlying parametric equations, and the coefficients are up to 1% accurate. On the other hand, Std-SINDy is not able to identify the correct form of the model. Furthermore, we compare the discovered model simulations using RK4-SINDy with ground truth beyond the training regime of the parameter $\mu$ in figure 11c,d. It exposes the strength of the parsimonious and interpretable discovered models.

## 6. Discussion

This work has introduced a compelling approach (RK4-SINDy) to discover nonlinear differential equations. For this, we have blended sparsity-promoting identification with a numerical integration scheme, namely, the classical fourth-order Runge–Kutta method. We note that the

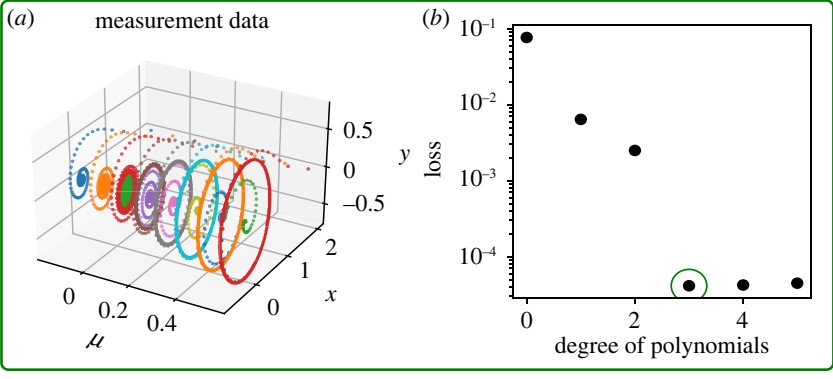

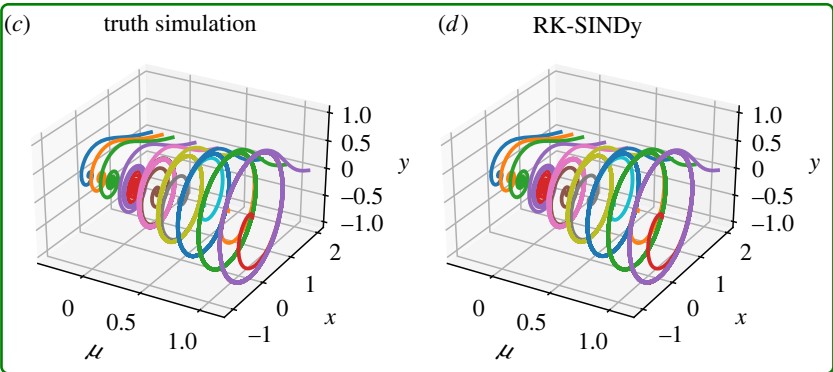

**Figure 11.** Hopf normal form: (*a*) the noisy measurements that are obtained using various parameter $\mu$. To identify correct degree polynomial basis in the dictionary, we do an assessment test, indicating degree-3 polynomials are sufficient to describe the dynamics (*b*). (*c,d*) A comparison of simulations of the ground truth model and identified models for the parameter $\mu$, illustrating the capability of generalizing beyond the training parameter regime. (Online version in colour.)

RK4 scheme could easily be exchanged with other high-order explicit or adaptive-time integrators using those presented in [29], and similar results could be expected. The appeal of the proposed methodology is that we do not require derivative information at any stage, still discovering differential equations. Hence, the proposed algorithm differs from previously suggested sparsity-promoting identification methods in the literature in this aspect. Consequently, we expect RK4-SINDy to perform better under sparsely sampled and corrupted data. We have demonstrated the efficiency of the approach on a variety of examples, namely linear and nonlinear damped oscillators, a model describing neural dynamics, chaotic behaviour and parametric differential equations. We have accurately discovered the Fitz–Hugh Nagumo model that describes the activation and de-activation of neurons. We have also illustrated the identification of the Lorenz model and have shown that the dynamics of identified models are intact on an attractor as it is more important for chaotic dynamics. The example of Michaelis–Menten kinetics highlights that the proposed algorithm can discover models that are given by a fraction of two functions. The example also shows the power of determining parsimonious models—that is, their generalization beyond the region in which data are selected. Furthermore, we have demonstrated the robustness of the proposed RK4-SINDy algorithm to sparsely sampled and corrupted measurement data. In the case of large noise, a noise-reduction filter such as Savitzky–Golay helps to improve the quality of the discovered governing equations. We have also reported a comparison with the sparse identification approach [15] and have observed the out-performance of RK4-SINDy over the latter approach.

This work opens many exciting doors for further research from both theory and practical perspectives. Since the approach aims at selecting the correct features from a dictionary containing a high-dimensional nonlinear feature basis, the construction of these feature bases in a dictionary plays a significant role in determining the success of the approach. There is no straightforward answer to this obstacle; however, there is some expectation that meaningful features may be constructed with the help of experts and empirical knowledge, or at least they may be realized in raw forms by them. Furthermore, we have solved the optimization problem (2.9) using a gradient-based method. We have observed that if feature functions in the dictionary are similar for given data, the convergence is slow, and sometimes it fails and is stuck in a non-sparse local minimum. In other words, the coherency between the feature functions is low. Hence, there is a need for the normalization step. In subsections (c) and (d), we have employed a normalization step to improve coherency. However, it is worth investigating better-suited strategies to normalize either data or feature spaces as a pre-processing step so that sparsity in the feature space remains intact. In addition to these, a thorough study on the performance of various noise-reduction methods (e.g. [35,52]) would provide deep insights into their appropriateness to RK4-SINDy, despite that we noted a good performance of the Sabitzky–Golay filter to reduced noise in our results. Moreover, one can also perform a statistical analysis by obtaining an ensemble of the sparse models using RK4-SINDy as done in [43].

Methods discovering interpretable models that generalize well beyond the training regime are limited, and the proposed method RK4-SINDy is among these. Additionally, approaches to discovering governing equations that also obey physical laws are even more rare. A very recent paper [53] has stressed that learning models can be made even more efficient by incorporating the laws of nature in the course of discovering equations, and the work [54] shows how physical constraints or empirical knowledge can be incorporated into SINDy. A solid example comes from discovering biological networks that often follow the mass-conversation law. Therefore, integrating physical laws in the course of discovering models will hopefully shape the future of finding explainable and generalizable differential equations.

Data accessibility. Our code and data can be found in the following link: https://github.com/mpimd-csc/RK4-SinDy.

Authors' contributions. P.G.: conceptualization, data curation, investigation, methodology, software, writing—original draft, writing—review and editing; P.B.: conceptualization, data curation, investigation, methodology, writing—review and editing.

All authors gave final approval for publication and agreed to be held accountable for the work performed therein.

Conflict of interest declaration. We declare we have no competing interest.

Funding. Both authors received partial support from BiGmax (www.bigmax.mpg.de/), the Max Planck Society's Research Network on Big-Data-Driven Materials Science. P.B. also obtained partial support from the German Research Foundation (DFG) Research Training Group 2297 'MathCoRe', Magdeburg, Germany.

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
