## [Peer Review File · Proceedings. Mathematical, Physical, and Engineering Sciences]

Review History

RSPA-2021-0883.R0 (Original submission)

Review form: Referee 1

Is the manuscript an original and important contribution to its field?

Excellent

Is the paper of sufficient general interest?

Excellent

Is the overall quality of the paper suitable?

Excellent

Can the paper be shortened without overall detriment to the main message?

Yes

Do you think some of the material would be more appropriate as an electronic appendix?

No

Do you have any ethical concerns with this paper?

No

Recommendation?

Accept with minor revision (please list in comments)

Comments to the Author(s)

I would like to support publication of the manuscript by Goyal and Benner. The paper is well written and thorough, with lots of good examples demonstrating their method and an open source package for reproducing the results.

I would ask they consider the following remarks which I feel are quite reasonable.

Some minor comments first to fix up

- Abstract: "given dictionary" to "given a dictionary"
- Paragraph 1: "Pareto front" not "Pareto font"
- There are probably more of these little things

A more important point is that there are three papers in particular I would like to point out that I think deserve more attention in the introduction since they are directly relevant to the current work and should be more appropriately highlighted given how they all are working very closely around the theme of imposing time-stepping constraints, for instance using RK4.

The first to do model discovery in this manner was Kevrekidis and co-workers in the late 1990s. The two papers below were quite ahead of their time in training neural networks around learning time-stepping and system ID. In addition, paper [33] should be highlighted instead of being buried in a list. Rudy et al [33] does something very similar in nature and it just seems that the two papers below and Rudy et al should merit deeper consideration since they have also very much worked around the constraint of time-steppers in neural network training. I believe that what is done in this work is great and I strongly support it, but it does seem to gloss over works that are the most similar, which seems quite unnecessary. Perhaps an additional paragraph in the introduction is warranted to highlight how time-stepping constraints have been used previously (perhaps even including ideas from neuralODE)

```
@article{gonzalez1998identification,
  title={Identification of distributed parameter systems: A neural net based approach},
  author={Gonzalez-Garcia, Raul and Rico-Martinez, Ramiro and Kevrekidis, Ioannis G},
  journal={Computers & chemical engineering},
  volume={22},
  pages={S965--S968},
  year={1998},
  publisher={Elsevier}
}
```

```
@inproceedings{rico1994continuous,
  title={Continuous-time nonlinear signal processing: a neural network based approach for gray box identification},
  author={Rico-Martinez, R and Anderson, JS and Kevrekidis, IG},
  booktitle={Proceedings of IEEE Workshop on Neural Networks for Signal Processing},
  pages={596--605},
  year={1994},
```

```
organization={IEEE}
}
```

Recently, Zhang and Schaeffer have shown the convergence of iterative thresholding. It seems it would be worth citing this work here since so much of what is done depends on the success of this thresholding.

```
@article{zhang2019convergence,
  title={On the convergence of the SINDy algorithm},
  author={Zhang, Linan and Schaeffer, Hayden},
  journal={Multiscale Modeling \& Simulation},
  volume={17},
  number={3},
  pages={948--972},
  year={2019},
  publisher={SIAM}
}
```

As for noise, the authors have correctly pointed out that derivative estimates are the most critical aspect to successful model identification. I would like to see a better reporting of what signal-to-noise level was used and were breakdown of the RK4-SINDy occurs as well. Standard SINDy has issues with noise, but since 2016, a number of variants have been derived that significantly outperform the original standard SINDy, so perhaps this is not the best comparison. Included in this denoising is the work of Rudy et al [33] which uses similar time-stepping constraints in order to separate signal from noise. SINDy can then be used on the remaining cleaned up data. But more recently, other techniques have been used which often simply rely on ensembling (see referenes below). This provides a major performance gain. A performance gain that can be probably leveraged also by RK4-SINDy.

```
@article{delahunt2021toolkit,
  title={A toolkit for data-driven discovery of governing equations in high-noise regimes},
  author={Delahunt, Charles B and Kutz, J Nathan},
  journal={arXiv preprint arXiv:2111.04870},
  year={2021}
}
```

```
@article{fasel2021ensemble,
  title={Ensemble-SINDy: Robust sparse model discovery in the low-data, high-noise limit, with active learning and control},
  author={Fasel, Urban and Kutz, J Nathan and Brunton, Bingni W and Brunton, Steven L},
  journal={arXiv preprint arXiv:2111.10992},
  year={2021}
}
```

Review form: Referee 2

Is the manuscript an original and important contribution to its field?

Acceptable

Is the paper of sufficient general interest?

Acceptable

Is the overall quality of the paper suitable?

Acceptable

Can the paper be shortened without overall detriment to the main message?

Yes

Do you think some of the material would be more appropriate as an electronic appendix?

No

Do you have any ethical concerns with this paper?

No

Recommendation?

Major revision is needed (please make suggestions in comments)

Comments to the Author(s)

The manuscript titled "Discovery of Nonlinear Dynamical Systems using a Runge-Kutta Inspired Dictionary-based Sparse Regression Approach" describes a combination of two ideas:

(a) Runge-Kutta template for an optimization cost, and (b) sparsity-promoting algorithms for the selection of a few functions out of a large dictionary.

The authors describe these ideas and their combination, provide two sparsity-promoting algorithms for function selection, and then compare their approach to the SINDy method in a list of computational experiments, including system identification with noisy observation, large time steps, and parameterized systems.

The idea to combine SINDy with a RK4-templated cost function is appealing, I have not seen this combination in other papers before. The fact that rational functions are also discussed is very interesting as well, since most dictionary based methods do not include them. The code provided in the repository is clear and well documented, which is very much appreciated.

Unfortunately, the current manuscript falls short in several categories that all need to be addressed before I can recommend publication. These are:

1) citation of the proper related work, 2) in-depths discussion of the advantage of a dictionary-based approach compared to "black-box" models, and 3) discussion of the challenges and shortcomings of the presented approach.

Let me address all three separately:

To 1)

The idea of a loss / cost function that is templated to RK4 (or other numerical integrators) is not new. The papers the authors cite in that direction are recent iterations of the idea, I have included many others from much earlier (1990s'). These manuscripts already described the benefits the authors mention, and more: robustness to noise, derivative-free system identification of parameterized systems, but also gray-box identification, PDEs, system identification for experimental data, bifurcation analysis, etc. The authors should not only cite these earlier works, but also contrast their approach to them and discuss in detail why a sparse dictionary is "better" than a neural network based one (also see point 2).

The papers I list go way beyond the computational experiments on simple ODE models in few dimensions the authors present, so the authors should argue why presenting simpler examples is enough.

To 2)

It is not at all clear why learning a few functions out of a pre-defined dictionary is "better" or

"more interpretable" than the "black-box" models, as the authors call them (I am guessing they mean neural networks? What about Gaussian processes with problem-adapted kernels, sparse kernel-SVM, etc.?). I know that the literature surrounding SINDy usually just equates "few terms in the equations" with "interpretable model", but there is no discussion as to why this is so (not in the SINDy paper, or in this manuscript): why is a model with a few terms out of a large dictionary "interpretable"? It is certainly parsimonious (by construction), but since the equations are not derived from certain principles but just "fit the data well", I do not see why one can interpret them better than any other black-box model. The interpretability problem is mostly hidden, because the authors (and work surrounding SINDy) *start* with interpretable models that already are given with a few terms, and then "discover" them again. But what happens if the approach is applied to data where there is no interpretable model available, and the approach yields 5-10 terms per coordinate? How would the authors "interpret" the result? What happens if two lists of functions fit the data (almost) equally well, which one is more "interpretable"? This is not discussed.

To 3)

The manuscript currently follows a common pattern in the machine learning-related literature: (a) describe an idea, (b) present an algorithm, (c) evaluate the performance on a few simple examples (preferably against another algorithm with inferior performance). This general approach is fine, and I appreciate the list of examples that nicely demonstrate different aspects of the identification procedure (parameters, different time steps, rational functions). However, in its current form, there is a severe lack of theoretical analysis of the algorithm, to a degree that is not acceptable for publication. It is understandable that not all theoretical aspects of a new algorithm are included in a new paper, but currently are almost none, the only real evaluation is through a few computational experiments.

Questions like the following should be addressed (if not with proofs, then at least with a discussion):

- how sensitive is the approach to noise, why did the authors choose the particular noise values and not more/other noise? More importantly: *Why* is the approach robust to noise, as illustrated with the experiments?
- if the data is NOT generated with a RK4 integration method (but e.g. with a step-size controlled version, or with a less/more accurate version), in what way does the learned ODE differ from the original ODE?
- what happens if the dictionary does not contain functions that are in the original model that generated the data? This is crucial to know for real data where the dictionary is always "wrong". It is also crucial to discuss how the approach generalizes in that case, because I do not think it would do much better than other approaches (why would it?).
- how does the approach scale when the number of variables in the ODE increases significantly, e.g. in a PDE discretization? How does one choose a proper dictionary, if even choosing third-order cross terms is infeasible because of the explosion of terms?
- why do the algorithms (1 and 2) converge? At what rate? The current description of algorithm 1 certainly does not always converge, because "setting $\Theta[\text{idx}]=0$ " is missing in it (even though it is mentioned in the description).
- for the approximation of rational functions, there is no discussion what happens if the denominator is (close to) zero. Does the convergence rate suffer? Does the algorithm break down?
- in the computational experiments with large time steps, it is not very clear that the "Std-SINDy" approach is using time derivatives that are approximated from the data (finite differencing?), which is the main reason (I guess?) that the results are so bad. What would happen if the data is noisy or with large time gaps, but the derivatives at the data points are exact?

Some minor issues in the text remain, I just list a few because the authors should focus on the big picture first:

- "2.9" should be "2.10" most of the time when referencing the optimization

- "gradient field" (page 6) should be "vector field", because it may not be a gradient of a potential
- "We refer to the proposed approach as Runge-Kutta inspired sparse identification" should probably also include "... of dynamics", otherwise it would just be "RK4-SIN"

Decision letter (RSPA-2021-0883.R0)

14-Feb-2022

Dear Dr Goyal

The Editor of Proceedings A has now received comments from referees on the above paper and would like you to revise it in accordance with their suggestions which can be found below (not including confidential reports to the Editor).

Please submit a copy of your revised paper within four weeks - if we do not hear from you within this time then it will be assumed that the paper has been withdrawn. In exceptional circumstances, extensions may be possible if agreed with the Editorial Office in advance.

Please note that it is the editorial policy of Proceedings A to offer authors one round of revision in which to address changes requested by referees. If the revisions are not considered satisfactory by the Editor, then the paper will be rejected, and not considered further for publication by the journal. In the event that the author chooses not to address a referee's comments, and no scientific justification is included in their cover letter for this omission, it is at the discretion of the Editor whether to continue considering the manuscript.

To revise your manuscript, log into <http://mc.manuscriptcentral.com/prsa> and enter your Author Centre, where you will find your manuscript title listed under "Manuscripts with Decisions." Under "Actions," click on "Create a Revision." Your manuscript number has been appended to denote a revision.

You will be unable to make your revisions on the originally submitted version of the manuscript. Instead, revise your manuscript and upload a new version through your Author Centre.

When submitting your revised manuscript, you will be able to respond to the comments made by the referee(s) and upload a file "Response to Referees" in Step 1: "View and Respond to Decision Letter". Please provide a point-by-point response to the comments raised by the reviewers and the editor(s). A thorough response to these points will help us to assess your revision quickly. You can also upload a 'tracked changes' version either as part of the 'Response to reviews' or as a 'Main document'.

IMPORTANT: Your original files are available to you when you upload your revised manuscript. Please delete any unnecessary previous files before uploading your revised version.

When revising your paper please ensure that it remains under 28 pages long. In addition, any pages over 20 will be subject to a charge (£150 + VAT (where applicable) per page). Your paper has been ESTIMATED to be 23 pages.

Open Access

You are invited to opt for open access, our author pays publishing model. Payment of open access fees will enable your article to be made freely available via the Royal Society website as

soon as it is ready for publication. For more information about open access please visit <https://royalsociety.org/journals/authors/open-access/>. The open access fee for this journal is £1700/\$2380/€2040 per article. VAT will be charged where applicable. Please note that if the corresponding author is at an institution that is part of a Read and Publishing deal you are required to select this option. See <https://royalsociety.org/journals/librarians/purchasing/read-and-publish/read-publish-agreements/> for further details.

Once again, thank you for submitting your manuscript to Proc. R. Soc. A and I look forward to receiving your revision. If you have any questions at all, please do not hesitate to get in touch.

Yours sincerely
 Raminder Shergill
 proceedingsa@royalsociety.org

Reviewer(s)' Comments to Author:

Referee: 1

Comments to the Author(s)

I would like to support publication of the manuscript by Goyal and Benner. The paper is well written and thorough, with lots of good examples demonstrating their method and an open source package for reproducing the results.

I would ask they consider the following remarks which I feel are quite reasonable.

Some minor comments first to fix up

- Abstract: "given dictionary" to "given a dictionary"
- Paragraph 1: "Pareto front" not "Pareto font"
- There are probably more of these little things

A more important point is that there are three papers in particular I would like to point out that I think deserve more attention in the introduction since they are directly relevant to the current work and should be more appropriately highlighted given how they all are working very closely around the theme of imposing time-stepping constraints, for instance using RK4.

The first to do model discovery in this manner was Kevrekidis and co-workers in the late 1990s.

The two papers below were quite ahead of their time in training neural networks around learning time-stepping and system ID. In addition, paper [33] should be highlighted instead of being buried in a list. Rudy et al [33] does something very similar in nature and it just seems that the two papers below and Rudy et al should merit deeper consideration since they have also very much worked around the constraint of time-steppers in neural network training. I believe that what is done in this work is great and I strongly support it, but it does seem to gloss over works that are the most similar, which seems quite unnecessary. Perhaps an additional paragraph in the introduction is warranted to highlight how time-stepping constraints have been used previously (perhaps even including ideas from neuralODE)

```
@article{gonzalez1998identification,
  title={Identification of distributed parameter systems: A neural net based approach},
  author={Gonzalez-Garcia, Raul and Rico-Martinez, Ramiro and Kevrekidis, Ioannis G},
  journal={Computers & chemical engineering},
  volume={22},
```

```

pages={S965--S968},
year={1998},
publisher={Elsevier}
}

```

```

@inproceedings{rico1994continuous,
title={Continuous-time nonlinear signal processing: a neural network based approach for gray box identification},
author={Rico-Martinez, R and Anderson, JS and Kevrekidis, IG},
booktitle={Proceedings of IEEE Workshop on Neural Networks for Signal Processing},
pages={596--605},
year={1994},
organization={IEEE}
}

```

Recently, Zhang and Schaeffer have shown the convergence of iterative thresholding. It seems it would be worth citing this work here since so much of what is done depends on the success of this thresholding.

```

@article{zhang2019convergence,
title={On the convergence of the SINDy algorithm},
author={Zhang, Linan and Schaeffer, Hayden},
journal={Multiscale Modeling \& Simulation},
volume={17},
number={3},
pages={948--972},
year={2019},
publisher={SIAM}
}

```

As for noise, the authors have correctly pointed out that derivative estimates are the most critical aspect to successful model identification. I would like to see a better reporting of what signal-to-noise level was used and were breakdown of the RK4-SINDy occurs as well. Standard SINDy has issues with noise, but since 2016, a number of variants have been derived that significantly outperform the original standard SINDy, so perhaps this is not the best comparison. Included in this denoising is the work of Rudy et al [33] which uses similar time-stepping constraints in order to separate signal from noise. SINDy can then be used on the remaining cleaned up data. But more recently, other techniques have been used which often simply rely on ensembling (see referenes below). This provides a major performance gain. A performance gain that can be probably leveraged also by RK4-SINDy.

```

@article{delahunt2021toolkit,
title={A toolkit for data-driven discovery of governing equations in high-noise regimes},
author={Delahunt, Charles B and Kutz, J Nathan},
journal={arXiv preprint arXiv:2111.04870},
year={2021}
}

```

```

@article{fasel2021ensemble,
title={Ensemble-SINDy: Robust sparse model discovery in the low-data, high-noise limit, with active learning and control},
author={Fasel, Urban and Kutz, J Nathan and Brunton, Bingni W and Brunton, Steven L},
journal={arXiv preprint arXiv:2111.10992},

```

```
year={2021}
}
```

Referee: 2

Comments to the Author(s)

The manuscript titled "Discovery of Nonlinear Dynamical Systems using a Runge-Kutta Inspired Dictionary-based Sparse Regression Approach" describes a combination of two ideas:

(a) Runge-Kutta template for an optimization cost, and (b) sparsity-promoting algorithms for the selection of a few functions out of a large dictionary.

The authors describe these ideas and their combination, provide two sparsity-promoting algorithms for function selection, and then compare their approach to the SINDy method in a list of computational experiments, including system identification with noisy observation, large time steps, and parameterized systems.

The idea to combine SINDy with a RK4-templated cost function is appealing, I have not seen this combination in other papers before. The fact that rational functions are also discussed is very interesting as well, since most dictionary based methods do not include them. The code provided in the repository is clear and well documented, which is very much appreciated.

Unfortunately, the current manuscript falls short in several categories that all need to be addressed before I can recommend publication. These are:

1) citation of the proper related work, 2) in-depths discussion of the advantage of a dictionary-based approach compared to "black-box" models, and 3) discussion of the challenges and shortcomings of the presented approach.

Let me address all three separately:

To 1)

The idea of a loss / cost function that is templated to RK4 (or other numerical integrators) is not new. The papers the authors cite in that direction are recent iterations of the idea, I have included many others from much earlier (1990s'). These manuscripts already described the benefits the authors mention, and more: robustness to noise, derivative-free system identification of parameterized systems, but also gray-box identification, PDEs, system identification for experimental data, bifurcation analysis, etc. The authors should not only cite these earlier works, but also contrast their approach to them and discuss in detail why a sparse dictionary is "better" than a neural network based one (also see point 2).

The papers I list go way beyond the computational experiments on simple ODE models in few dimensions the authors present, so the authors should argue why presenting simpler examples is enough.

To 2)

It is not at all clear why learning a few functions out of a pre-defined dictionary is "better" or "more interpretable" than the "black-box" models, as the authors call them (I am guessing they mean neural networks? What about Gaussian processes with problem-adapted kernels, sparse kernel-SVM, etc.?). I know that the literature surrounding SINDy usually just equates "few terms in the equations" with "interpretable model", but there is no discussion as to why this is so (not in the SINDy paper, or in this manuscript): why is a model with a few terms out of a large dictionary "interpretable"? It is certainly parsimonious (by construction), but since the equations are not derived from certain principles but just "fit the data well", I do not see why one can interpret them better than any other black-box model. The interpretability problem is mostly hidden, because the authors (and work surrounding SINDy) *start* with interpretable models that already are given with a few terms, and then "discover" them again. But what happens if the approach is applied to data where there is no interpretable model available, and the approach

yields 5-10 terms per coordinate? How would the authors "interpret" the result? What happens if two lists of functions fit the data (almost) equally well, which one is more "interpretable"? This is not discussed.

To 3)

The manuscript currently follows a common pattern in the machine learning-related literature: (a) describe an idea, (b) present an algorithm, (c) evaluate the performance on a few simple examples (preferably against another algorithm with inferior performance). This general approach is fine, and I appreciate the list of examples that nicely demonstrate different aspects of the identification procedure (parameters, different time steps, rational functions). However, in its current form, there is a severe lack of theoretical analysis of the algorithm, to a degree that is not acceptable for publication. It is understandable that not all theoretical aspects of a new algorithm are included in a new paper, but currently are almost none, the only real evaluation is through a few computational experiments.

Questions like the following should be addressed (if not with proofs, then at least with a discussion):

- how sensitive is the approach to noise, why did the authors choose the particular noise values and not more/other noise? More importantly: *Why* is the approach robust to noise, as illustrated with the experiments?
- if the data is NOT generated with a RK4 integration method (but e.g. with a step-size controlled version, or with a less/more accurate version), in what way does the learned ODE differ from the original ODE?
- what happens if the dictionary does not contain functions that are in the original model that generated the data? This is crucial to know for real data where the dictionary is always "wrong". It is also crucial to discuss how the approach generalizes in that case, because I do not think it would do much better than other approaches (why would it?).
- how does the approach scale when the number of variables in the ODE increases significantly, e.g. in a PDE discretization? How does one choose a proper dictionary, if even choosing third-order cross terms is infeasible because of the explosion of terms?
- why do the algorithms (1 and 2) converge? At what rate? The current description of algorithm 1 certainly does not always converge, because "setting $\Theta_{\text{idx}}=0$ " is missing in it (even though it is mentioned in the description).
- for the approximation of rational functions, there is no discussion what happens if the denominator is (close to) zero. Does the convergence rate suffer? Does the algorithm break down?
- in the computational experiments with large time steps, it is not very clear that the "Std-SINDy" approach is using time derivatives that are approximated from the data (finite differencing?), which is the main reason (I guess?) that the results are so bad. What would happen if the data is noisy or with large time gaps, but the derivatives at the data points are exact?

Some minor issues in the text remain, I just list a few because the authors should focus on the big picture first:

- "2.9" should be "2.10" most of the time when referencing the optimization
- "gradient field" (page 6) should be "vector field", because it may not be a gradient of a potential
- "We refer to the proposed approach as Runge-Kutta inspired sparse identification" should probably also include "... of dynamics", otherwise it would just be "RK4-SIN"

Author's Response to Decision Letter for (RSPA-2021-0883.R0)

See Appendix A.

RSPA-2021-0883.R1 (Revision)

Review form: Referee 1

Is the manuscript an original and important contribution to its field?

Excellent

Is the paper of sufficient general interest?

Excellent

Is the overall quality of the paper suitable?

Excellent

Can the paper be shortened without overall detriment to the main message?

Yes

Do you think some of the material would be more appropriate as an electronic appendix?

No

Do you have any ethical concerns with this paper?

No

Recommendation?

Accept as is

Comments to the Author(s)

The authors did a very nice job with their updated manuscript

Review form: Referee 2

Is the manuscript an original and important contribution to its field?

Excellent

Is the paper of sufficient general interest?

Good

Is the overall quality of the paper suitable?

Good

Can the paper be shortened without overall detriment to the main message?

Yes

Do you think some of the material would be more appropriate as an electronic appendix?

No

Do you have any ethical concerns with this paper?

No

Recommendation?

Accept with minor revision (please list in comments)

Comments to the Author(s)

I appreciate the comments and changes in the manuscript. Two of my main concerns were about the theoretical analyses of the algorithms, which the authors now discuss briefly and cite appropriate sources, and the additional literature, which is also covered now. I believe the manuscript contains - and already contained in the last iteration - an important and also rather immediate contribution to system identification, combining the ideas of integrator templates with sparse identification. I believe the authors can easily correct the remaining issues I have with its current state. Unfortunately, I cannot explain them very briefly, so I had to write a longer text again.

1) My biggest issue with the current manuscript (not the author response) is still about the claim of interpretability. The author response did not contain a literature list, so I had to search in the manuscript for the paper [Brunton et al, 2016], which the authors claim "... provides many examples of why obtaining analytical expressions for dynamical models can be essential, including interpretability and generalizability."

I assume the authors either reference

"Discovering governing equations from data by sparse identification of nonlinear dynamical systems"

or

"Sparse Identification of Nonlinear Dynamics with Control (SINDYc)".

Both papers do not mention the term "interpretability" once, they only mention "parsimonious" models - which of course is true by construction (sparsity).

My issue is that parsimonious models are not inherently interpretable. If the authors of this manuscript claim they are, they have to argue why that is, not just in the author response to me, but also in the manuscript. As an example: why is a model such as $f(x) = A \cdot \tanh(B \cdot x + b) + a$ (i.e., a neural network with one layer) less interpretable than a model like $f(x) = \frac{k(x) + g(x)}{1 + h(x)}$ (equation 4.6 in the paper)? To be very clear: I am not saying the network is interpretable, far from it. I just want to point out that claiming interpretability is not something that should be taken lightly, and has to be backed up in the text. The author write that "based on discussions with our colleagues from biotechnology and chemical engineering departments, that selecting terms form a dictionary of common functions facilitates the interpretation of the identified model", which is exactly what I would like to see mentioned in the manuscript. Why did the colleagues see it this way? Why do they think a model such as $\dot{x} = \frac{\sin(x)}{1 + \tanh(x)} + \cos(\exp(x))^2$ is interpretable? Two or three sentences would be enough.

2) The point that one can use deep learning (Champion et al., PNAS, 2019) to map to a latent space where the dictionary can be applied would then take away the entire interpretability claim again, I suppose, so probably that should not be followed up in the manuscript.

3) I do not think my comments about the denominator being zero (or close to it) were clear. If the denominator is zero, the response of the model for \dot{x} is undefined. I was asking what the large numbers that occur if it is close to zero will do to the sparsity promoting algorithm. As of now, the manuscript has no discussion whatsoever about the denominator being exactly or close to zero. One sentence would be enough here, especially for the behavior at small values (exact zero values are very unlikely, numerically).

4) Some minor points:

* In Algorithm 1, it should be clarified what happens if the set ($|\Theta| < \lambda$) is empty, both in step 3 and step 7: how do you compute the error? I guess it is set to zero.

* Algorithm 2: "the constraint $\Theta(\text{small_idx})$." is missing an actual constraint (I guess " $\Theta(\text{small_idx})=0$ "?)

Decision letter (RSPA-2021-0883.R1)

22-Apr-2022

Dear Dr Goyal,

On behalf of the Editor, I am pleased to inform you that your Manuscript RSPA-2021-0883.R1 entitled "Discovery of Nonlinear Dynamical Systems using a Runge-Kutta Inspired Dictionary-based Sparse Regression Approach" has been accepted for publication subject to minor revisions in Proceedings A. Please find the referees' comments below.

The reviewer(s) have recommended publication, but also suggest some minor revisions to your manuscript. Therefore, I invite you to respond to the reviewer(s)' comments and revise your manuscript. Please note that we have a strict upper limit of 28 pages for each paper. Please endeavour to incorporate any revisions while keeping the paper within journal limits. Please note that page charges are made on all papers longer than 20 pages. If you cannot pay these charges you must reduce your paper to 20 pages before submitting your revision. Your paper has been ESTIMATED to be 24 pages. We cannot proceed with typesetting your paper without your agreement to meet page charges in full should the paper exceed 20 pages when typeset. If you have any questions, please do get in touch.

It is a condition of publication that you submit the revised version of your manuscript within 7 days. If you do not think you will be able to meet this date please let me know in advance of the due date.

To revise your manuscript, log into <https://mc.manuscriptcentral.com/prsa> and enter your Author Centre, where you will find your manuscript title listed under "Manuscripts with Decisions." Under "Actions," click on "Create a Revision." Your manuscript number has been appended to denote a revision.

You will be unable to make your revisions on the originally submitted version of the manuscript. Instead, revise your manuscript and upload a new version through your Author Centre.

When submitting your revised manuscript, you will be able to respond to the comments made by the referee(s) and upload a file "Response to Referees" in Step 1: "View and Respond to Decision Letter". Please provide a point-by-point response to the comments raised by the reviewers and the editor(s). A thorough response to these points will help us to assess your revision quickly. You can also upload a 'tracked changes' version either as part of the 'Response to reviews' or as a 'Main document'.

IMPORTANT: Your original files are available to you when you upload your revised manuscript. Please delete any redundant files before completing the submission process.

When uploading your revised files, please make sure that you include the following as we cannot proceed without these:

- 1) A text file of the manuscript (doc, txt, rtf or tex), including the references, tables (including captions) and figure captions. Please remove any tracked changes from the text before submission. PDF files are not an accepted format for the "Main Document".
- 2) A separate electronic file of each figure (tif, eps or print-quality pdf preferred). The format should be produced directly from original creation package, or original software format.
- 3) Electronic Supplementary Material (ESM): all supplementary materials accompanying an accepted article will be treated as in their final form. Note that the Royal Society will not edit or typeset supplementary material and it will be hosted as provided. Please ensure that the supplementary material includes the paper details where possible (authors, article title, journal name). Supplementary files will be published alongside the paper on the journal website and posted on the online figshare repository (<https://figshare.com>). The heading and legend provided for each supplementary file during the submission process will be used to create the figshare page, so please ensure these are accurate and informative so that your files can be found in searches. Files on figshare will be made available approximately one week before the accompanying article so that the supplementary material can be attributed a unique DOI. Alternatively you may upload a zip folder containing all source files for your manuscript as described above with a PDF as your "Main Document". This should be the full paper as it appears when compiled from the individual files supplied in the zip folder.

Article Funder

Please ensure you fill in the Article Funder question on page 2 to ensure the correct data is collected for FundRef (<http://www.crossref.org/fundref/>).

Media summary

Please ensure you include a short non-technical summary (up to 100 words) of the key findings/importance of your paper. This will be used for to promote your work and marketing purposes (e.g. press releases). The summary should be prepared using the following guidelines:

- *Write simple English: this is intended for the general public. Please explain any essential technical terms in a short and simple manner.
- *Describe (a) the study (b) its key findings and (c) its implications.
- *State why this work is newsworthy, be concise and do not overstate (true 'breakthroughs' are a rarity).
- *Ensure that you include valid contact details for the lead author (institutional address, email address, telephone number).

Cover images

We welcome submissions of images for possible use on the cover of Proceedings A. Images should be square in dimension and please ensure that you obtain all relevant copyright permissions before submitting the image to us. If you would like to submit an image for consideration please send your image to proceedingsa@royalsociety.org

Open Access

You are invited to opt for open access, our author pays publishing model. Payment of open access fees will enable your article to be made freely available via the Royal Society website as soon as it is ready for publication. For more information about open access please visit

<https://royalsociety.org/journals/authors/open-access/>. The open access fee for this journal is £1700/\$2380/€2040 per article. VAT will be charged where applicable. Please note that if the corresponding author is at an institution that is part of a Read and Publishing deal you are required to select this option. See <https://royalsociety.org/journals/librarians/purchasing/read-and-publish/read-publish-agreements/> for further details.

Once again, thank you for submitting your manuscript to Proceedings A and I look forward to receiving your revision. If you have any questions at all, please do not hesitate to get in touch.

Best wishes
 Raminder Shergill
 proceedingsa@royalsociety.org
 Proceedings A

Reviewer(s)' Comments to Author:

Referee: 2

Comments to the Author(s)

I appreciate the comments and changes in the manuscript. Two of my main concerns were about the theoretical analyses of the algorithms, which the authors now discuss briefly and cite appropriate sources, and the additional literature, which is also covered now. I believe the manuscript contains - and already contained in the last iteration - an important and also rather immediate contribution to system identification, combining the ideas of integrator templates with sparse identification. I believe the authors can easily correct the remaining issues I have with its current state. Unfortunately, I cannot explain them very briefly, so I had to write a longer text again.

1) My biggest issue with the current manuscript (not the author response) is still about the claim of interpretability. The author response did not contain a literature list, so I had to search in the manuscript for the paper [Brunton et al, 2016], which the authors claim "... provides many examples of why obtaining analytical expressions for dynamical models can be essential, including interpretability and generalizability."

I assume the authors either reference

"Discovering governing equations from data by sparse identification of nonlinear dynamical systems"

or

"Sparse Identification of Nonlinear Dynamics with Control (SINDYc)".

Both papers do not mention the term "interpretability" once, they only mention "parsimonious" models - which of course is true by construction (sparsity).

My issue is that parsimonious models are not inherently interpretable. If the authors of this manuscript claim they are, they have to argue why that is, not just in the author response to me, but also in the manuscript. As an example: why is a model such as $f(x) = A \cdot \tanh(B \cdot x + b) + a$ (i.e., a neural network with one layer) less interpretable than a model like $f(x) = \frac{k(x) + g(x)}{1 + h(x)}$ (equation 4.6 in the paper)? To be very clear: I am not saying the network is interpretable, far from it. I just want to point out that claiming interpretability is not something that should be taken lightly, and has to be backed up in the text. The author write that "based on discussions with our colleagues from biotechnology and chemical engineering departments, that selecting terms form a dictionary of common functions facilitates the interpretation of the identified model", which is exactly what I would like to see mentioned in the manuscript. Why did the colleagues see it this way? Why do they think a model such as

" $\dot{x} = \sin(x)/(1+\tanh(x)) + \cos(\exp(x))^2$ " is interpretable? Two or three sentences would be enough.

2) The point that one can use deep learning (Champion et al., PNAS, 2019) to map to a latent space where the dictionary can be applied would then take away the entire interpretability claim again, I suppose, so probably that should not be followed up in the manuscript.

3) I do not think my comments about the denominator being zero (or close to it) were clear. If the denominator is zero, the response of the model for \dot{x} is undefined. I was asking what the large numbers that occur if it is close to zero will do to the sparsity promoting algorithm. As of now, the manuscript has no discussion whatsoever about the denominator being exactly or close to zero. One sentence would be enough here, especially for the behavior at small values (exact zero values are very unlikely, numerically).

4) Some minor points:

* In Algorithm 1, it should be clarified what happens if the set $(|\Theta| * \text{Algorithm 2: "the constraint } \Theta(\text{small_idx})."$ is missing an actual constraint (I guess " $\Theta(\text{small_idx})=0$ ")?

Referee: 1

Comments to the Author(s)

The authors did a very nice job with their updated manuscript

Author's Response to Decision Letter for (RSPA-2021-0883.R1)

See Appendix B.

Decision letter (RSPA-2021-0883.R2)

Dear Colleagues,

I am writing to inform you that the Editor has made a decision on manuscript entitled "Discovery of Nonlinear Dynamical Systems using a Runge-Kutta Inspired Dictionary-based Sparse Regression Approach" which you kindly refereed for Proceedings A. Please find the authors' decision letter below.

On behalf of the Editor of Proceedings A, we thank you for your help with this article and we look forward to your input in the future.

Decision made on this manuscript: Accept as is

Best wishes

Raminder Shergill

proceedingsa@royalsociety.org

@@date to be populated upon sending@@

Dear Dr Goyal

On behalf of the Editor, I am pleased to inform you that your manuscript entitled "Discovery of Nonlinear Dynamical Systems using a Runge-Kutta Inspired Dictionary-based Sparse Regression Approach" has been accepted in its final form for publication in Proceedings A.

Our Production Office will be in contact with you in due course. You can expect to receive a proof of your article soon. Please contact the office to let us know if you are likely to be away from e-mail in the near future. If you do not notify us and comments are not received within 5 days of sending the proof, we may publish the paper as it stands.

As a reminder, you have provided the following 'Data accessibility statement' (if applicable). Please remember to make any data sets live prior to publication, and update any links as needed when you receive a proof to check. It is good practice to also add data sets to your reference list. Statement (if applicable): Our code and data can be found in the following link: <https://github.com/mpimd-csc/RK4-SinDy>.

Open access

You are invited to opt for open access, our author pays publishing model. Payment of open access fees will enable your article to be made freely available via the Royal Society website as soon as it is ready for publication. For more information about open access please visit <https://royalsociety.org/journals/authors/which-journal/open-access/>. The open access fee for this journal is £1700/\$2380/€2040 per article. VAT will be charged where applicable.

Note that if you have opted for open access then payment will be required before the article is published – payment instructions will follow shortly. If you wish to opt for open access then please inform the editorial office (proceedingsa@royalsociety.org) as soon as possible.

Your article has been estimated as being 24 pages long. Our Production Office will inform you of the exact length at the proof stage.

Proceedings A levies charges for articles which exceed 20 printed pages. (based upon approximately 540 words or 2 figures per page). Articles exceeding this limit will incur page charges of £150 per page or part page, plus VAT (where applicable).

Under the terms of our licence to publish you may post the author generated postprint (ie. your accepted version not the final typeset version) of your manuscript at any time and this can be made freely available. Postprints can be deposited on a personal or institutional website, or a recognised server/repository. Please note however, that the reporting of postprints is subject to a media embargo, and that the status the manuscript should be made clear. Upon publication of the definitive version on the publisher's site, full details and a link should be added.

You can cite the article in advance of publication using its DOI. The DOI will take the form: 10.1098/rspa.XXXX.YYYY, where XXXX and YYYY are the last 8 digits of your manuscript number (eg. if your manuscript number is RSPA-2017-1234 the DOI would be 10.1098/rspa.2017.1234).

For tips on promoting your accepted paper see our blog post: <https://royalsociety.org/blog/2020/07/promoting-your-latest-paper-and-tracking-your-results/>

Thank you for your submission. On behalf of the Editors of the journal, we look forward to your continued contributions to the Journal.

Best wishes

Raminder Shergill,
Proceedings A Editorial Office
proceedingsa@royalsociety.org

Appendix A

Response to referee' reports on the submission “Discovery of Nonlinear Dynamical Systems using a Runge-Kutta Inspired Dictionary-based Sparse Regression Approach”

Pawan Goyal and Peter Benner

We would like to thank the handling editor and the reviewers for spending their time reading and evaluating our article. We are grateful for their feedback, comments, and criticisms, which, we believe, helped us to improve the paper. We have made changes in the revised version of the paper to address the reviewers' comments. In what follows, we discuss each point made by the reviewers in detail and explain the modifications made in the latest version of the paper.

1 Response to Reviewer 1

We would like to thank Reviewer #1 for their nice words. Thank you for several suggestions to improve the paper. We have taken care of the mentioned minor comments, and in the following, we respond to the major comments.

- A more important point is that there are three papers in particular I would like to point out that I think deserve more attention in the introduction since they are directly relevant to the current work and should be more appropriately highlighted given how they all are working very closely around the theme of imposing time-stepping constraints, for instance using RK4.

The first to do model discovery in this manner was Kevrekidis and co-workers in the late 1990s. The two papers below were quite ahead of their time in training neural networks around learning time-stepping and system ID. In addition, paper [33] should be highlighted instead of being buried in a list. Rudy et al [33] does something very similar in nature and it just seems that the two papers below and Rudy et al should merit deeper consideration since they have also very much worked around the constraint of time-steppers in neural network training. I believe that what is done in this work is great and I strongly support it, but it does seem to gloss over works that are the most similar, which seems quite unnecessary. Perhaps an additional paragraph in the introduction is warranted to highlight how time-stepping constraints have been used previously (perhaps even including ideas from neuralODE)

@article{gonzalez1998identification,
title=Identification of distributed parameter systems: A neural net based approach,
author=González-García, Raul and Rico-Martínez, Ramiro and Kevrekidis, Ioannis G,
journal=Computers & chemical engineering,
volume=22,
pages=S965-S968,
year=1998,

```
publisher=Elsevier
}
```

```
@inproceedings{rico1994continuous,
title=Continuous-time nonlinear signal processing: a neural network based approach for gray box identification,
author=Rico-Martinez, R and Anderson, JS and Kevrekidis, IG,
booktitle=Proceedings of IEEE Workshop on Neural Networks for Signal Processing,
pages=596-605,
year=1994,
organization=IEEE
}
```

Ans: Thank you for the suggestion. We are really sorry that some of this work slipped our attention. We have added a separate paragraph discussing all these work including our own recent work that can be seen as extension of the Rudy et al. work in which an estimate of the noise explicitly is not needed, and can also handle missing irregular data easily.

- Recently, Zhang and Schaeffer have shown the convergence of iterative thresholding. It seems it would be worth citing this work here since so much of what is done depends on the success of this thresholding.

```
@article{zhang2019convergence,
title=On the convergence of the SINDy algorithm,
author=Zhang, Linan and Schaeffer, Hayden,
journal=Multiscale Modeling & Simulation,
volume=17,
number=3,
pages=948-972,
year=2019,
publisher=SIAM
}
```

We have added a discussion about the paper and have also added a remark about convergence.

- As for noise, the authors have correctly pointed out that derivative estimates are the most critical aspect to successful model identification. I would like to see a better reporting of what signal-to-noise level was used and were breakdown of the RK4-SINDY occurs as well. Standard SINDY has issues with noise, but since 2016, a number of variants have been derived that significantly outperform the original standard SINDY, so perhaps this is not the best comparison. Included in this denoising is the work of Rudy et al [33] which uses similar time-stepping constraints in order to separate signal from noise. SINDy can then be used on the remaining cleaned up data. But more recently, other techniques have been used which often simply rely on ensembling (see referenes below). This provides a major performance gain. A performance gain that can be probably leveraged also by RK4-SINDy.

```
@article{delahunt2021toolkit,
title=A toolkit for data-driven discovery of governing equations in high-noise regimes,
author=Delahunt, Charles B and Kutz, J Nathan,
journal=arXiv preprint arXiv:2111.04870,
year=2021
}
```

```
@article{fasel2021ensemble,
title=Ensemble-SINDy: Robust sparse model discovery in the low-data, high-noise limit, with active learning and control,
author=Fasel, Urban and Kutz, J Nathan and Brunton, Bingni W and Brunton, Steven L,
journal=arXiv preprint arXiv:2111.10992,
```

```
year=2021
}
```

First, we would like to note that we submitted the paper in November 2021; we were not aware of these two manuscripts that seem to have appeared around the time when we finished our submission. In the revised paper, we have now reported the SNR (see Table 2). We had shown in our earlier version the breakdown of RK4-SINDy, but the breaking down is much later than for classical SINDy. Moreover, in the revised version, we have added a higher level of noise than used before where RK4-SINDy breaks down, as well as standard SINDy. Moreover, we have added a larger time-step for the collected data in the first example, where RK-SINDy also breaks down. In summary, RK4-SINDy would also break down if the noise level is very high or the sampling time for data collection is large, but the breakdown occurs much later as compared to the standard SINDy.

Indeed, neural network-assisted methods exist to remove the noise from the signal, like [Rudy et al., 2018] and our recent work [Goyal/Benner, 2021]. Indeed, it can improve the performance of standard SINDy to learn sparse dynamical systems, but at the same time, RK4-SINDy can also benefit from it as pointed out by the review. Moreover, note that neural network-assisted methods will not obliterate the noise from the signal, and we have observed that RK4-SINDy is more robust to the noise to learn dynamical systems. Therefore, even if we remove most of the noise using the aforementioned methods, we expect the RK4-SINDy method to recover a model closer to the truth.

Furthermore, you rightly pointed to the recent work on Ensemble-SINDy—which has got our attention as well—to obtain statistics about the coefficients for the features in the dictionary. As you also mentioned, the methods such as an ensemble scheme using bootstrapping can also be applied in RK4-SINDy and obtain statistics like in Ensemble-SINDy. So, the performance of the RK4-SINDy can also be improved accordingly. In principle, every additional mechanism that brings some improvement to standard SINDy will bring an analogous improvement to RK4-SINDy as well. We have mentioned the ensemble approach in the conclusion of the revised paper, but we refrain from adding any new results because it will bring a whole new discussion and methods to discuss in the paper, which diverts from the main theme of the paper.

2 Response to Reviewer 2

We would like to thank Reviewer #2. Thank you for the suggestions to improve the papers. In the following, we discuss your comments in detail.

- The idea of a loss/cost function that is templated to RK4 (or other numerical integrators) is not new. The papers the authors cite in that direction are recent iterations of the idea, I have included many others from much earlier (1990s'). These manuscripts already described the benefits the authors mention, and more: robustness to noise, derivative-free system identification of parameterized systems, but also gray-box identification, PDEs, system identification for experimental data, bifurcation analysis, etc. The authors should not only cite these earlier works, but also contrast their approach to them and discuss in detail why a sparse dictionary is "better" than a neural network based one (also see point 2). The papers I list go way beyond the computational experiments on simple ODE models in few dimensions the authors present, so the authors should argue why presenting simpler examples is enough.

Ans: Thank you for the detailed comment. We have added a paragraph where we have mentioned the earlier work from the 1990ies discussing the idea of involving numerical integrators in learning dynamical models. Here, we would like to stress that the main advantage of the sparse regression approaches are the resulting analytical expressions for the identified dynamical models. Detailed discussions of the

benefits can be found in several papers by Brunton, Kutz, and co-authors. We do not claim that the SINDy approach ultimately outperforms other machine learning techniques, but see it as alternative direction in the vast literature on nonlinear system identifications, that is worth being further explored and developed. An advantage as compared to neural network-based approaches is that the training phase might be cheaper (both computational and data-requirement), but this also requires more and deeper investigations that are beyond the scope of this paper. The paper [Brunton et al. 2016] provides many examples of why obtaining analytical expressions for dynamical models can be essential, including interpretability and generalizability. These models tend to generalize better as compared to neural network-based approaches outside the training regime. In our example of Michaelis-Menten kinetics, we have illustrated that when we can obtain an analytical expression describing the underlying dynamics, we can extrapolate, which is not possible using neural network-based modeling. Moreover, interpretability can be gained using analytical expressions, which is not possible using neural network-based approaches. For example, consider a biological process in which four species interact with each other. Using an analytical model, we can identify not only which species are interacting with which, but also we can determine how they influence each other. It will not be easily possible using neural network-based models. Therefore, there are several attempts to obtain analytical expressions describing dynamics. Symbolic regression or genetic algorithms can also be thought of as a step towards obtaining sparse models.

- It is not at all clear why learning a few functions out of a pre-defined dictionary is "better" or "more interpretable" than the "black-box" models, as the authors call them (I am guessing they mean neural networks? What about Gaussian processes with problem-adapted kernels, sparse kernel-SVM, etc.?). I know that the literature surrounding SINDy usually just equates "few terms in the equations" with "interpretable model", but there is no discussion as to why this is so (not in the SINDy paper, or in this manuscript): why is a model with a few terms out of a large dictionary "interpretable"? It is certainly parsimonious (by construction), but since the equations are not derived from certain principles but just "fit the data well", I do not see why one can interpret them better than any other black-box model. The interpretability problem is mostly hidden, because the authors (and work surrounding SINDy) *start* with interpretable models that already are given with a few terms, and then "discover" them again. But what happens if the approach is applied to data where there is no interpretable model available, and the approach yields 5-10 terms per coordinate? How would the authors "interpret" the result? What happens if two lists of functions fit the data (almost) equally well, which one is more "interpretable"? This is not discussed.

Ans: We believe, and this is also based on discussions with our colleagues from biotechnology and chemical engineering departments, that selecting terms form a dictionary of common functions facilitates the interpretation of the identified model. E.g., many known kinetic terms in chemical reactions are of polynomial and rational nature. So if we include such terms in the dictionary, an expert in biochemistry or chemical engineering might be able to identify an unknown reaction mechanism that would be hard to observe from a neural network or kernel-based representation of the dynamics. Then, of course, this approach has to prove its merits in practice, and should be compared and analyzed against other approaches as mentioned by you. We plan to do this in close collaboration, e.g., with the biotechnology department of our Max Planck Institute, but we feel that the sparse symbolic regression approach first needs to be developed methodologically to be mature enough for working with real wetlab data. Our paper is one steps towards reaching this goal, and as said above, we believe that the approach warrants these efforts!

Indeed, the success of the whole dictionary-based modeling lies in the fact that the dictionary is rich enough so that a few terms can explain the dynamics. This is one of the necessary assumptions in this philosophy.

It is not necessary that we can always obtain sparse models using SINDy, especially when the dictio-

nary does not include the right features. But at the same time, failure of identifying a parsimonious model and having a large regression error may guide the redesign the features in the dictionary.

However, we also would like to mention that there are works that utilize the power of deep learning to identify the correct coordinate system in which dynamics can be determined by sparse regression, see [Champion et al., PNAS, 2019 (<https://doi.org/10.1073/pnas.1906995116>)]. Another way to look at this approach can be to look for a coordinate transformation using deep learning whose dynamics can be given by choosing a few terms from the constructed dictionary. This is one way to circumvent the problem when one cannot build an appropriate dictionary.

If two lists of functions fit the data almost equally, one should use a model that is explained using minimum functions from the dictionary. This goes along the line of Occam’s Razor.

Naturally, we have started with models where the ground truth is known to illustrate the methodology. This is a very typical when a new methodology is being proposed. If we target only one particular application starting from collecting data, it will target a particular audience, whereas we here aim to provide a general tool. Indeed, in our future projects, we apply these tools to particular applications coming from biology processes and robotics.

In the end, we would like to mention that sparse identification of dynamical systems provides “a tool” based on the assumption that a few key features from a huge dictionary of functions can explain the underlying dynamics. And in this paper, we proposed a scheme to avoid taking time-derivate of data to obtain continuous dynamical models using a numerical integration scheme. A new section is devoted to identifying rational functions that often appear in identifying biological models.

We also mention that one can also include certain principles or constraints—known empirically or from an expert—in learning dynamical models using SINDy, see [Loiseau/Brunton, 2018].

- The manuscript currently follows a common pattern in the machine learning-related literature: (a) describe an idea, (b) present an algorithm, (c) evaluate the performance on a few simple examples (preferably against another algorithm with inferior performance). This general approach is fine, and I appreciate the list of examples that nicely demonstrate different aspects of the identification procedure (parameters, different time steps, rational functions). However, in its current form, there is a severe lack of theoretical analysis of the algorithm, to a degree that is not acceptable for publication. It is understandable that not all theoretical aspects of a new algorithm are included in a new paper, but currently are almost none, the only real evaluation is through a few computational experiments.

We recap that in this paper, we have discussed how to integrate a numerical integration scheme for identifying dynamical models using sparse regression, which has not been done in the past. We also discussed a way to deal with identifying systems containing rational non-linearity. We have provided two algorithms to solve the resulting non-convex optimization problem, seeking a sparse solution as well. However, a complete analysis such as convergence of the algorithm is beyond the scope of this paper, although we totally agree that it is a very relevant topic but require a complete study on its own. In the revised paper, we have mentioned the recent article about the convergence analysis of the standard SINDy thresholding algorithm. A similar study for our proposed algorithms remains open as future research topics which certainly would be quite involved. This was also a big challenge for the thresholding algorithm in [Brunton et al., 2016] even though the underlying optimization problem was linear and convex.

However, we note that the proposed algorithms will always terminate as either the number of indices set to zero is increased (which terminates when the dictionary is exhausted) or the error criterion is satisfied. But the question remains whether the algorithm will converge to the correct sparse solution. As stated above, a thorough study requires further research. However, a remedy to this can be to use the rationale of an ensemble, recently proposed by [Fasel et al. <https://arxiv.org/abs/2111.10992>]. We can build an ensemble of sparse models which can provide statistical quantities for the feature candidates in the dictionary, which can be used to construct a final sparse model based on statistical

tools such as the p -values. In the revised paper, we have discussed this; see p.7.

- Questions like the following should be addressed (if not with proofs, then at least with a discussion):

- how sensitive is the approach to noise, why did the authors choose the particular noise values and not more/other noise? More importantly: *Why* is the approach robust to noise, as illustrated with the experiments?

Ans: For the first example, we had already done a comprehensive study concerning noise by varying the noise level in data (see Table 2). Due to the limitation of pages, we did not report a similar study for the remaining four examples by varying noise levels and time differences in measurements due to page limits. Therefore, for the other examples, we have chosen a single noise level just for illustration and comparison purpose. There is no particular reason except for the page-limit and repetition of the discussions. We have discussed a potential reason why our proposed method is robust. One main reason is that we reduce the error of approximating time-derivatives. The Runge-Kutta step is an integration step, and as known, doing integration involving noisy data is more robust than taking derivative involving noisy data. The second potential is that the governing equations for each dependent variable are determined simultaneously, in contrast to the standard SINDy, where an equation of each dependent variable is obtained separately though while doing a numerical integration, they interact.

- if the data is NOT generated with a RK4 integration method (but e.g. with a step-size controlled version, or with a less/more accurate version), in what way does the learned ODE differ from the original ODE?

Ans: Please note that the data are NOT generated with a RK4 integration method in our experiments. They are rather generated using the Python function `solve_ivp` that uses an adaptive time stepping to meet a desired tolerance. We have used the default settings in `scipy` library. We have given a `time grid` as an argument to obtain values at the given time grid. Thus, there is a discrepancy in the discovered model and the truth model, though small. We have tried to clarify now in the beginning of the numerical section how the data are collected in the experiments.

- what happens if the dictionary does not contain functions that are in the original model that generated the data? This is crucial to know for real data where the dictionary is always "wrong". It is also crucial to discuss how the approach generalizes in that case, because I do not think it would do much better than other approaches (why would it?).

Ans: As we mentioned earlier, the success of the proposed approach or any SINDy-inspired methods lies on the fact that a huge dictionary contains right candidate feature functions; otherwise, it will fail. It has been discussed in the appendix of the paper by [Brunton et al., PNAS 2016]. However, when a potential candidate is missing from the dictionary but the dictionary contains polynomial features, we observe that the identified model is very close to a Taylor series expansion, see page 23 of the appendix of the paper by [Brunton et al., PNAS 2016] (Link: <https://www.pnas.org/action/downloadSupplement?doi=10.1073%2Fpnas.1517384113&file=pnas.1517384113.sapp.pdf>). We also observe the same for RK4-SINDy. We do not claim that this issue that already is present in the standard SINDy can be solved by our method.

Lastly, we again would like to mention that there are works that utilize the power of deep learning to identify the correct coordinate system in which dynamics can be determined by sparse regression, see [Champion et al., PNAS, 2019 (<https://doi.org/10.1073/pnas.1906995116>)]. Another way to look at this approach can be to look for a coordinate transformation using deep learning whose dynamics can be given by choosing a few terms from the constructed dictionary. This is one way to circumvent the problem when one cannot build an appropriate dictionary.

- how does the approach scale when the number of variables in the ODE increases significantly, e.g.

in a PDE discretization? How does one choose a proper dictionary, if even choosing third-order cross terms is infeasible because of the explosion of terms?

Ans: First of all, the proposed methods are for ODEs which do not have co-relation at least spatially. If one has data obtained from e.g., PDE discretization, then one should not apply the proposed methodology directly. Instead, one should first apply a dimensionality reduction method (e.g., using principle component analysis) to obtain a low-dimensional representation or embedding, and then use our approach to learn a model in low-dimensional embeddings.

- why do the algorithms (1 and 2) converge? At what rate? The current description of algorithm 1 certainly does not always converge, because "setting Theta[idx]=0" is missing in it (even though it is mentioned in the description).

Ans: The convergence of Algorithm 1 heavily depends on the algorithm used to the underlying optimization problem in equation (2.10). In our paper, we used the classical gradient descent algorithm and implemented it with the help of PyTorch. It guarantees convergence to at least local minima, providing a sufficiently small learning rate. However, choosing an advanced optimization algorithm, particularly second-order methods, can improve convergence. So, it boils down to solving the optimization problem in equation (2.10). In this paper, our purpose is not to study the performance of different optimization algorithms to solve the underlying optimization problem. We intend to study the performance of the different optimization algorithms and observe convergence in our future work . In this paper, we clarify how we have solved the optimization problem; see the numerical section on page 11. However, we point out that a major question remains whether the thresholding algorithm converges to the correct sparse solution since it depends on the thresholding parameter; this is yet to be answered.

We apologize for the omission — now we have added in Step 5 of Algorithm 1.

- for the approximation of rational functions, there is no discussion what happens if the denominator is (close to) zero. Does the convergence rate suffer? Does the algorithm break down?

Note that when the denominator is zero, then the dynamics will change rapidly. We firmly believe that it does not have any effect as long as there are sufficiently many points capturing the dynamical behavior and a suitable normalization of the data are used. We performed a normalization step in our examples (see, e.g., the example of Michaelis-Menten kinetics). However, it also remains to study how the normalization of data affects the overall performance of the algorithms. We have discussed this point in our conclusions and will do further investigation in this direction.

- in the computational experiments with large time steps, it is not very clear that the "Std-SINDy" approach is using time derivatives that are approximated from the data (finite differencing?), which is the main reason (I guess?) that the results are so bad. What would happen if the data is noisy or with large time gaps, but the derivatives at the data points are exact?

Ans: Indeed, for Std-SINDy, we approximate the derivative information using finite-difference when data are noiseless, whereas, in RK4-SINDy, we do not require to estimate the time-derivatives directly. If the derivative information is exactly known, then the whole motivation of using RK4-SINDy in the first place does not apply. In this case, one can directly use the standard SINDy. However, applications where the derivative is known exactly, and measurements are noisy or with large time gaps are rare. We cannot think of any real-world application where we will know the derivative information precisely but not the measurements. The work of RK4-SINDy starts with the motivation that we do not have time-derivation information, and data can be sparsely sampled.

We have also modified the manuscript based on your minor comments.

3 Conclusion

We once again thank the associate editor and all referees for their valuable remarks and grateful to them. We hope that our answers have adequately addressed all the elements raised by the reviewers.

Appendix B

Response to referee' reports on the submission “Discovery of Nonlinear Dynamical Systems using a Runge-Kutta Inspired Dictionary-based Sparse Regression Approach”

Pawan Goyal and Peter Benner

We would again like to thank the handling editor and the reviewers for spending their time reading and evaluating our article. We have made changes in the revised version of the paper to address Reviewer#2 comments. In what follows, we discuss each point made by him/her in detail.

1 Response to Reviewer 2

We discuss each of the points below.

1. My biggest issue with the current manuscript (not the author response) is still about the claim of interpretability. The author response did not contain a literature list, so I had to search in the manuscript for the paper [Brunton et al, 2016], which the authors claim "... provides many examples of why obtaining analytical expressions for dynamical models can be essential, including interpretability and generalizability."

I assume the authors either reference "Discovering governing equations from data by sparse identification of nonlinear dynamical systems" or "Sparse Identification of Nonlinear Dynamics with Control (SINDYc)".

Both papers do not mention the term "interpretability" once, they only mention "parsimonious" models - which of course is true by construction (sparsity).

My issue is that parsimonious models are not inherently interpretable. If the authors of this manuscript claim they are, they have to argue why that is, not just in the author response to me, but also in the manuscript. As an example: why is a model such as

$$f(x) = A * \tanh(B * x + b) + a$$

(i.e., a neural network with one layer) less interpretable than a model like

$$f(x) = k(x) + g(x)/(1 + h(x))$$

(equation 4.6 in the paper)? To be very clear: I am not saying the network is interpretable, far from it. I just want to point out that claiming interpretability is not something that should be taken lightly, and has to be backed up in the text. The author write that "based on discussions with our colleagues from biotechnology and chemical engineering departments, that selecting terms form a dictionary of common functions facilitates the interpretation of the identified model", which is exactly what I would

like to see mentioned in the manuscript. Why did the colleagues see it this way? Why do they think a model such as " $\dot{x} = \sin(x)/(1 + \tanh(x)) + \cos(\exp(x))^2$ " is interpretable? Two or three sentences would be enough.

Ans: First, we would like to stress a point that we do not claim parsimonious implies interpretability or vice-versa. We meant here that parsimonious models might lead to interpretability more often than black-box models, e.g., neural networks.

The paper "Discovering governing equations from data by sparse identification of nonlinear dynamical systems" may not directly have a word *interpretability* in the paper. Still, if you look at the examples, particularly *PDE for Vortex Shedding Behind an Obstacle*, you notice how the learned model is interpretable. Fluid dynamics practitioners can extract important information about the dominant dynamical modes behind the obstacle. We have now used this as a motivating example in our introduction, citing this work, see p.2 and 2nd paragraph. Furthermore, we like to highlight that a parsimonious model in biology applications can expose the interconnection between the species, which is a step towards interpretability. Additionally, biological models often contain rational nonlinearities, which can also give an interpretation. For example, consider a system:

$$\dot{x} = -\frac{x}{1+x}. \quad (1)$$

The above equation indicates as the rate of change of x is linear for small x , and it is constant for large x . This information or interpretation directly follows from the model (1). To infer such information using neural network-based models even with a single layer will not be straightforward. We hope that we can convince that a parsimonious model may lead to interpretability with these two examples. However, we again stress that we do not claim that parsimonious means interpretability.

2. The point that one can use deep learning (Champion et al., PNAS, 2019) to map to a latent space where the dictionary can be applied would then take away the entire interpretability claim again, I suppose, so probably that should not be followed up in the manuscript.

Ans: Well, we disagree that learning a latent space where dynamics can be given by a few functions from a large dictionary completely takes away the interpretability claim. Following [Champion et al., PNAS, 2019], the Fourier transform was introduced to simplify the representation of the heat equations, resulting in sparse, diagonal, decoupled linear systems. In Champion et al., there is an example of a nonlinear pendulum where data are collected by capturing a video of the simulation that is in Cartesian coordinates. Still, there is an angular coordinate where the dynamics are sparse, i.e., $\ddot{\theta}(t) = \sin(\theta(t))$, and this model can be interpreted better than in Cartesian coordinates. Therefore, a coordinate transformation does not always remove interpretability. However, we agree that it will not always yield an interpretable model.

In any case, we have not followed up our discussion in the paper using [Champion et al., PNAS, 2019], as we are seeking to identify parsimonious models in those coordinates where data are collected.

3. I do not think my comments about the denominator being zero (or close to it) were clear. If the denominator is zero, the response of the model for \dot{x} is undefined. I was asking what the large numbers that occur if it is close to zero will do to the sparsity promoting algorithm. As of now, the manuscript has no discussion whatsoever about the denominator being exactly or close to zero. One sentence would be enough here, especially for the behavior at small values (exact zero values are very unlikely, numerically).

Ans: We have added a remark stating that there might be some numerical and optimization challenges in case the denominator is small, see p.10. This is potentially due to the large gradient; thus, one needs to do data collection and data normalization with care, although, in our example, we did not observe any expected behavior. Please also consider the typical type of rational function as in (1), where in biological models, $x(t)$ would be positive, so the the denominator is always greater than 1.

4. Some minor points:

* In Algorithm 1, it should be clarified what happens if the set $(|\Theta|$ * Algorithm 2: "the constraint $\Theta(\textit{small_idx})$." is missing an actual constraint (I guess " $\Theta(\textit{small_idx}) = 0$ "?)

Ans: We have corrected the mistake in Algorithm 2. You rightly said it should have been $\Theta(\textit{small_idx}) = 0$. But we did not understand the comment about Algorithm 1. It seems to be an incomplete statement.